# Transcription readthrough is prevalent in healthy human tissues and associated with inherent genomic features

Paulo Caldas [1,2✉], Mariana Luz [1,2], Simone Baseggio [1,2], Rita Andrade[1,2], Daniel Sobral [1,2,3] & Ana Rita Grosso [1,2✉]

Transcription termination is a crucial step in the production of conforming mRNAs and functional proteins. Under cellular stress conditions, the transcription machinery fails to identify the termination site and continues transcribing beyond gene boundaries, a phenomenon designated as transcription readthrough. However, the prevalence and impact of this phenomenon in healthy human tissues remain unexplored. Here, we assessed transcription readthrough in almost 3000 transcriptome profiles representing 23 human tissues and found that 34% of the expressed protein-coding genes produced readthrough transcripts. The production of readthrough transcripts was restricted in genomic regions with high transcriptional activity and was associated with inefficient splicing and increased chromatin accessibility in terminal regions. In addition, we showed that these transcripts contained several binding sites for the same miRNA, unravelling a potential role as miRNA sponges. Overall, this work provides evidence that transcription readthrough is pervasive and non-stochastic, not only in abnormal conditions but also in healthy tissues. This suggests a potential role for such transcripts in modulating normal cellular functions.

[1] Associate Laboratory i4HB - Institute for Health and Bioeconomy, NOVA School of Science and Technology, Universidade NOVA de Lisboa, 2829-516 Caparica, Portugal. [2] UCIBIO – Applied Molecular Biosciences Unit, Department of Life Sciences, NOVA School of Science and Technology, Universidade NOVA de Lisboa, 2829-516 Caparica, Portugal. [3]Present address: Genomics and Bioinformatics Unit, Department of Infectious Diseases, National Institute of Health Doutor Ricardo Jorge (INSA), Lisbon, Portugal. ✉email: p.caldas@fct.unl.pt; argrosso@fct.unl.pt

Transcription termination is a highly regulated process involving nascent transcript cleavage and RNA Polymerase II (RNAPII) release to produce conforming RNAs[1]. In certain conditions, the transcription machinery fails to recognize the transcription termination site and continues transcribing beyond annotated gene boundaries, a phenomenon termed transcription readthrough (TRT)[2–6]. Such aberrant transcription produces longer RNAs designated as readthrough transcripts (RT transcripts) or Downstream-of-Gene (DoG) containing transcripts[2]. These transcripts have been detected under a variety of cellular stress conditions, including hyperosmotic stress, heat shock, oxidative stress[2], hypoxia[7,8], viral infections[3,9] and even cancer[4]. Moreover, inactivation of modulators of histone modifications, splicing, or transcription termination can also produce aberrant transcripts[4,10–13].

These transcripts are typically described as RNA molecules extending at least 5 kbps past the normal termination site of their gene of origin (RT gene) and are retained in the nucleus close to their transcription site, most likely associated with chromatin[2,5,6]. In this context, it has been suggested that they may play a role in supporting chromatin structure. Readthrough transcription can also invade neighboring genes affecting their expression and impacting the transcriptome widely. For instance, downstream genes (usually silenced pseudogenes) can be transcribed without the activation of their promoters[14]. In contrast, functional anti-sense RNAs can be transcribed by the readthrough of convergent genes, resulting in the repression of host genes[15]. In addition, readthrough transcription can lead to the production of circular RNAs from downstream genes[9] or produce RNA chimeras by combining coding elements from different genes, where the intergenic regions and flanking exons (lacking splice sites) are removed by the splicing machinery[4].

Overall, transcription readthrough may severely modify the transcriptome and threaten the integrity of vital gene expression programs, but it may also produce functional noncoding RNAs with roles in gene regulation or cellular processes. However, the prevalence and functional impact of transcription readthrough in healthy tissues remains elusive.

Here, we assessed the levels of transcription readthrough (TRT) across 23 human tissues from 2778 high-throughput sequencing transcriptome (RNAseq) profiles available from the Genotype-Tissue Expression project[16–18]. We showed that TRT occurs frequently across several healthy human tissues, with higher expression levels contributing to exacerbating the effect. However, the expression levels can only partially explain such behavior, suggesting that certain gene features may play a role in propagating these termination defects. Our findings suggest that inefficient splicing, increased chromatin accessibility and depletion of CG-rich regions downstream of RT genes are some of these trademarks. Moreover, we found that readthrough transcripts contained several binding sites for the same miRNA, indicating a potential role as miRNA sponges. As a result, our work shows that transcription readthrough is not exclusive to pathological conditions but is pervasive in healthy tissues, suggesting a potential role for readthrough transcripts in the regulation of cellular processes.

## Results

**Transcription readthrough is pervasive in healthy tissues**. To assess the prevalence of transcription readthrough in normal cells, we analyzed the transcriptome profiles of 2778 samples from 23 different healthy human tissues obtained from the GTEx project[16–18] (Fig. 1a; Supplementary Fig. 1a; Supplementary Data 1). We used ARTDeco[19] to depict transcription termination defects by searching for significant read coverage downstream of the transcription termination site (TTS) of each expressed gene (see Methods for details) (Fig. 1b). Since GTEx samples were profiled using nonstranded RNAseq libraries, approximately 28% of the RT transcripts identified could correspond to downstream genes being expressed in the opposite strand (Supplementary Fig. 1b). To ensure a robust list of RT genes, we filtered out such false-positive cases by removing the RT transcripts overlapping genes in the opposite strand (being classified as undefined genes). However, this approach also eliminates legitimate RT transcripts with close downstream neighbors, resulting in an underestimation of the real transcriptional readthrough occurring in the cell. In fact, we applied the same approach to a small collection of strand-specific transcriptome profiles[20], confirming that some of these ambiguous cases were indeed valid transcription readthrough events (Supplementary Fig. 1b). Nevertheless, to take advantage of the extensive data available on the GTEx repository, we proceeded with the characterization of TRT in healthy tissues by discarding the dubious cases and considering only the underestimated but robust list of RT genes. To overcome the known effects of death and post-mortem cold ischemia on human tissue transcriptomes[21,22], we considered only adults with fast deaths and no terminal diseases (see Methods for details). Furthermore, we did not find any strong correlation between the levels of transcription readthrough and the attributes of the GTEx samples, such as ischemic time or RNA integrity number (Supplementary Data 2).

Our analysis confirmed the existence of transcription read-through across a variety of healthy tissues, where 10-20% of the expressed genes in each tissue produced RT transcripts (Fig. 1c; Supplementary Data 1). The number of RT genes varied within each tissue type (Supplementary Fig. 1c), which could be explained by the heterogeneity in tissue sampling. Notably, the large number of undefined cases in all the tissues suggests that this phenomenon may occur even more frequently. The brain cerebellum and testis showed the highest number of RT genes, whereas the lowest amount was found in the skeletal muscle and heart regions (Fig. 1c; Supplementary Fig. 1c). These findings suggest that transcription readthrough occurrence may be associated with cellular proliferation rates. However, no significant correlation was found between the expression of the proliferation marker KI67 and readthrough levels (Spearman R = 0.20, p value = 0.33, Supplementary Fig. 1d).

Interestingly, ~85% of the 7138 RT genes found across all tissues corresponded to protein-coding genes (Fig. 1d). This enrichment was significantly more pronounced relative to genes without readthrough (Fisher's exact test p value < 0.05; Supplementary Fig. 1e). In fact, 34% of the human protein-coding genes expressed in tissues (excluding protein-coding from undefined cases) produced RT transcripts, reinforcing the pervasiveness of transcription readthrough in normal cells. Moreover, our analysis revealed that 64% of RT transcripts were detected in at least two different tissues (Fig. 1e), with 20% being expressed in half of the tissues (Supplementary Fig. 1f). Consistent with this, physiologically related tissues showed similar RT patterns, with clustering resembling segregation when expressed genes were used (Supplementary Fig. 1g). Thus, these findings suggest that transcription readthrough is not a stochastic process and its occurrence may be associated with specific cellular or genomic aspects.

**Production of RT transcripts is restricted in regions with high gene activity**. To characterize the occurrence of transcriptional readthrough under normal cellular conditions, we explored several gene features. Overall, we found a moderate and significant correlation between the expression levels of the RT tail and the respective RT gene (Pearson R > 0.4, p value < 0.001; Fig. 2a, b). However, we did not observe a relevant increase in the expression

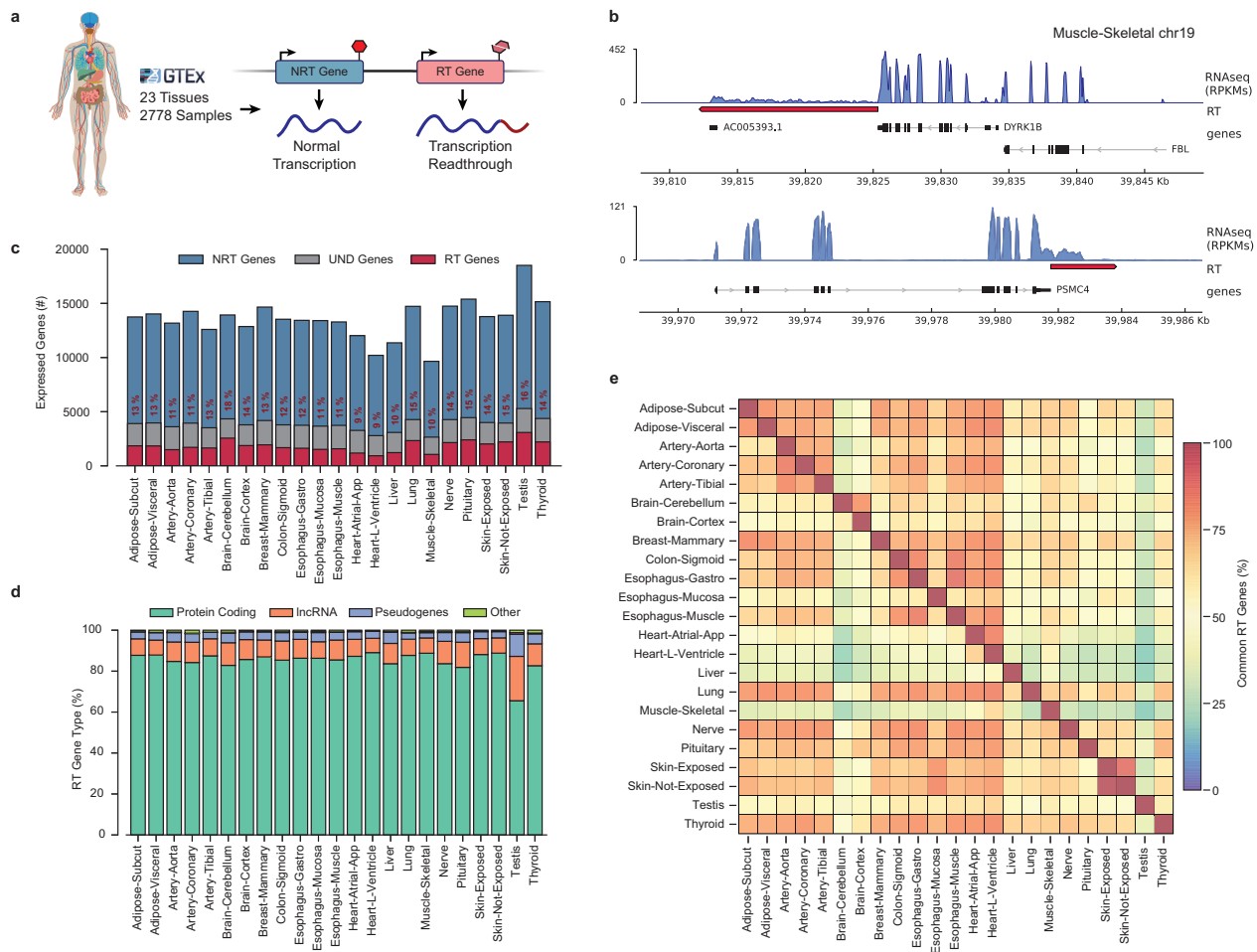

**Fig. 1 Transcription readthrough prevalence in healthy human tissues. a** Schematic representation of the overall approach for depicting readthrough genes (RT genes) and non-readthrough genes (NRT genes). Human illustration design by Vexels. **b** RNAseq profiles for genes *DYRK1B* and *PSMC4* showing the transcription readthrough region (red) in skeletal muscle. RNAseq coverage is represented as reads per kilobase per million mapped reads (RPKMs). Boxes represent exons separated by introns shown as solid lines. Assembly GRCh38, gencode annotation v37. **c** Expressed genes in each tissue classified according to our computational approach: RT genes (red), NRT genes (blue), and undefined (UND genes, gray). **d** Proportion of RT genes detected in each tissue grouped by gene type: protein-coding, lncRNA, pseudogenes, and other genes. **e** Heatmap representing the percentage of RT genes common between each tissue pair.

levels of RT genes when compared with genes without transcription readthrough (NRT) (Cohen's d < 0.25 and Mann-Whitney rank test FDR < 0.05, Supplementary Fig. 2a). In fact, high gene expression levels do not seem to be sufficient to produce RT transcripts, since the proportion of highly expressed genes without transcription readthrough was higher than 70% in most tissues (Fig. 2c). These findings are consistent with previous studies showing that RT transcripts are produced regardless of the transcriptional levels of their host genes[2,5,23].

Overall, the length of RT transcripts ranged from 2 kbp (minimum defined length by ARTDeco) up to 60 kbp, with an average size of 5 kbp across all tissues (Fig. 2d). However, we did not find any correlation between RT transcript length and expression levels of the respective RT genes (Spearman R = 0.035, Supplementary Fig. 2b). Given the long length of RT transcripts, we hypothesized that the closest presence of actively transcribed neighboring genes could influence the occurrence of transcription readthrough. We examined the distance between the termination site of each group of genes (RT versus NRT) and the downstream expressed gene, but we did not find any significant difference between the groups (Cohen's d < 0.02, Supplementary Fig. 2c).

We then investigated how transcription readthrough occurrence differed across chromosomes. We found a significant enrichment of RT genes on chromosome 4 and a significant depletion on chromosomes 16 and 17 in several tissues relative to all expressed genes (Fig. 2e). Importantly, no enrichment was found for genes without readthrough (Supplementary Fig. 2d). Such discrepancy could be associated with the different transcription activity across chromosomes. Indeed, we found a moderate but significant negative correlation between transcription readthrough occurrence and the density of expressed genes for each chromosome (Spearman's R = −0.657, p value < 0.01, Fig. 2f, Supplementary Data 3). Such a negative association persisted even when using the tissue strand-specific profiles, discarding any potential technical artifacts introduced by our filter (Spearman R = −0.472, p value < 0.05; Supplementary Fig. 2e). In addition, since gene density is not uniform along each chromosome, we confirmed these findings by considering small segments within each chromosome (Spearman R = −0.415, p value < 0.01 Supplementary Fig. 2f). Therefore, our results suggest that nearby high gene activity can restrict transcription readthrough.

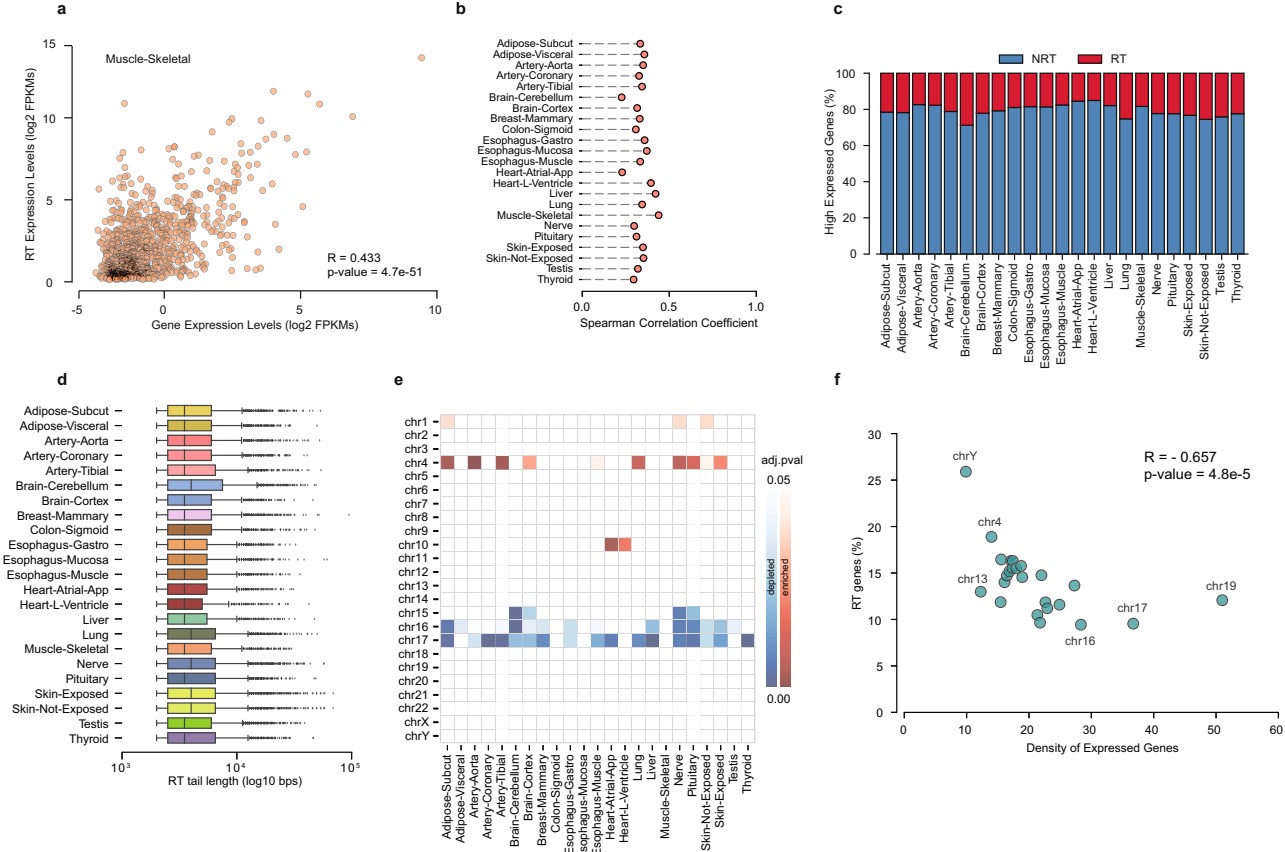

**Fig. 2 Genomic features of RT genes in healthy tissues. a** Scatterplot of expression levels of the RT region and respective RT genes in the skeletal muscle (log2 FPKMs). **b** Spearman Correlation Coefficients between the expression levels of the RT region and the respective RT genes for each tissue. **c** Proportion of highly expressed genes in each tissue with (RT genes) and without transcription readthrough (NRT genes). **d** Length of the RT transcript tail identified for each tissue. Boxplot whiskers indicate the 25th and 75th percentiles, while the dots represent all outliers skewing the distribution. **e** Heatmap representing the enrichment (red) or depletion (blue) of RT gene proportions across different chromosomes for each tissue. Color gradients represent Fisher's Exact test FDR. **f** Scatterplot RT transcripts percentage in all tissues and density of expressed genes for each chromosome.

**RT transcripts are delimited by specific epigenomic features.** Since gene expression is linked to chromatin accessibility[24], we characterized the chromatin landscape of readthrough transcribed regions using molecular profiles of healthy tissues from the Epigenome Roadmap Project[25]. After identifying RT and NRT genes in nine healthy tissues, we assessed the enrichment of 25 chromatin states encompassing 11 histone modifications and DNAse profiles (Supplementary Fig. 3a). Notably, the transcription termination site (TTS) vicinity of RT genes showed enrichment for chromatin states typical of 3' end transcribed regions and enhancers (chromatin states 9 and 11) (Fig. 3a). To deeply explore the epigenetic landscape associated with transcription readthrough, we applied a permutation approach with expression-matched genes for each mark associated with enriched chromatin states (Supplementary Fig. 3a–c; Methods for details). In fact, when comparing the expression-matched RT and NRT genes, we observed a higher presence of histone marks typical of transcription elongation (H3K36me3) at the 3' end of RT genes relative to genes without readthrough, which continued beyond the transcription termination site (Fig. 3b, c and Supplementary Fig. 3d). Some tissues also showed enrichment of accessible regulatory chromatin (H3K4me1 and H3K27ac) after the transcription termination site (Fig. 3b and Supplementary Fig. 3d). The presence of these three chromatin marks beyond the 3' end of RT genes has been previously reported under conditions of stress or HIV infection[5,6]. Thus, our findings show that such epigenetic alterations are also associated with the production of

transcription readthrough in healthy conditions. Furthermore, we also characterized the terminal regions of the RT transcripts, where we found enrichment for an enhancer-associated chromatin state (chromatin state 13; Fig. 3a). Consistently, the ends of RT transcripts showed higher levels of enhancer-specific histones H3K4me1, H3K27ac, and H3K4me2 (Fig. 3b, c; Supplementary Fig. 3d). Moreover, we detected high levels of DNase hypersensitivity profiling, indicating increased accessibility. This suggests that enhancer activity may prevent such aberrant elongation. Thus, our findings indicate that transcription readthrough in normal tissues is associated with specific chromatin states.

**Splicing efficiency and sequence termination elements influence Transcription Readthrough.** Gene expression is a complex cellular process with many closely coupled steps where each one functionally impacts the next[26]. Given that decreased co-transcriptional splicing efficiency may lead to termination defects[3], we explored the association between inefficient splicing and transcription readthrough in healthy human tissues. When comparing expression-matched genes, we observed intron retention levels higher in RT versus NRT genes for some tissues (e.g., liver, skeletal muscle and esophagus mucosa) (Fig. 4a, b and Supplementary Fig. 4a; Methods for details). Moreover, a significantly moderate correlation was found between the respective RT tail expressions (Supplementary Fig. 4b). However, these differences were not consistent across all tissues (Fig. 4b). The

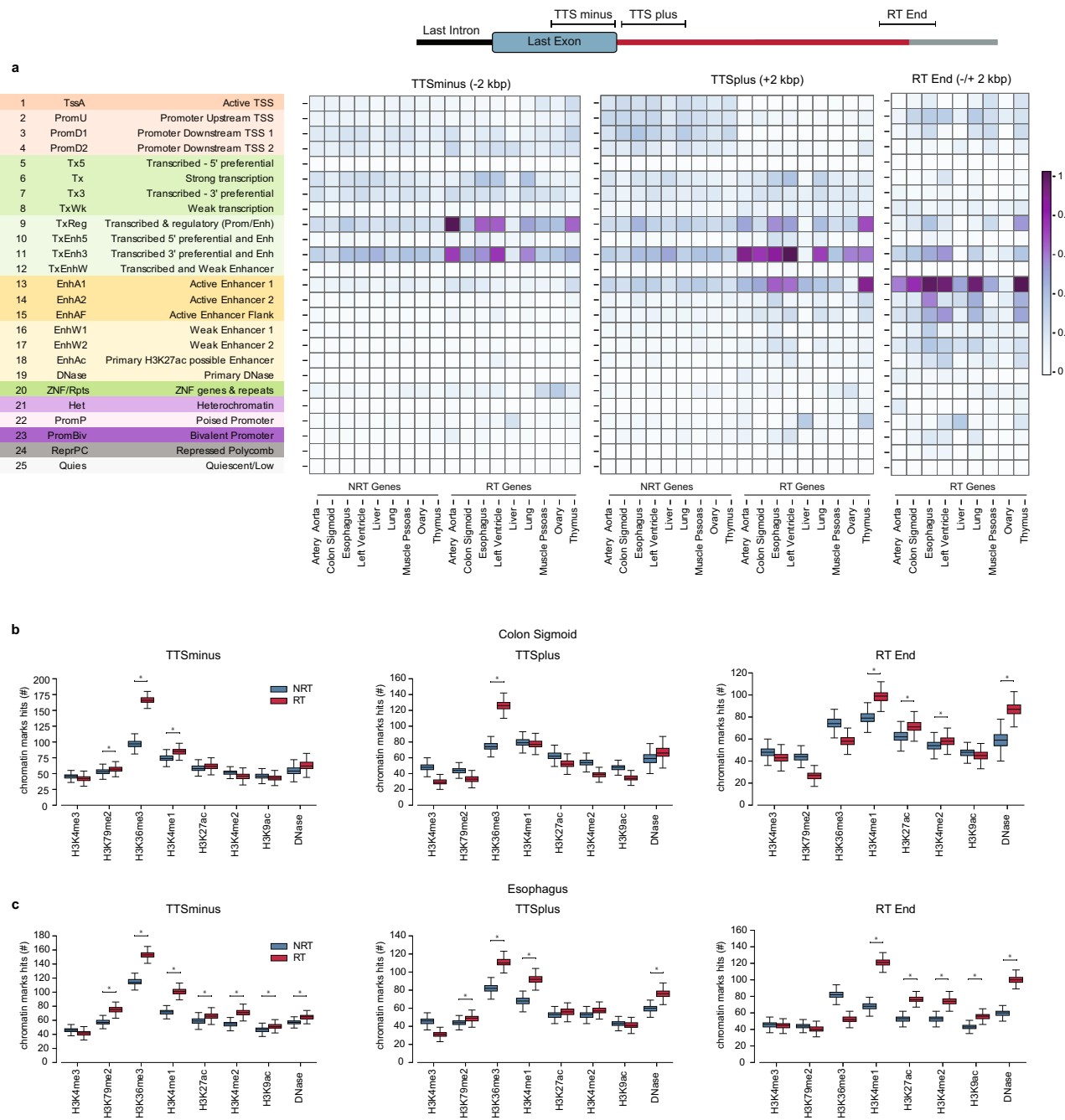

**Fig. 3 Epigenomic features of RT genes in healthy tissues. a** Heatmap representing the enrichment of 25 chromatin states for regions ($+/-$ 2Kbps) upstream Transcription Termination Site (TTSminus), downstream TTS (TTSplus) and around the terminal region of RT transcripts (RT End). **b**, **c** Number of RT/NRT genes and RT ends with chromatin marks associated with the enriched chromatin states in different regions for colon-sigmoid (**b**) and esophagus (**c**). Boxplot whiskers represent the 25th and 75th percentiles, while the mid-line in each box represents the median. The '*' indicates statistically significant differences: Mann-Whitney test $p$ value < 0.01; Cohen's d > 1.

retention of introns occurs frequently in both RT and NRT genes, especially in the last two introns (Fig. 4b: Supplementary Fig. 4a).

Notably, we found a striking association between transcription readthrough occurrence and the number of introns for all expressed genes (Spearman R = 0.82, $p$ value < 0.001; Fig. 4c, Supplementary Data 4). Intronless genes seemed to be less prone to readthrough (<5%), and this tendency increased with the total intron count, reaching a plateau where approximately 30% of genes with more than 20 introns produced RT transcripts. To further explore the potential cause-effect link between abnormal splicing patterns and transcription readthrough at the genome-

wide level, we reanalyzed the transcriptome profiles of human cells in which the knockout of the canonical splicing factor U2AF1 leads to global levels of intron retention[27]. We observed that the proportion of RT transcripts produced by each RT gene increased upon the inactivation of the splicing factor (Fig. 4d). All together, these findings suggest that intron number is correlated with splicing efficiency, which may influence the molecular events taking place at gene ends.

Besides accurate splicing, the presence of specific pause sites and a strong poly(A) signal downstream of genes can also modulate transcription termination[12]. Therefore, we decided to

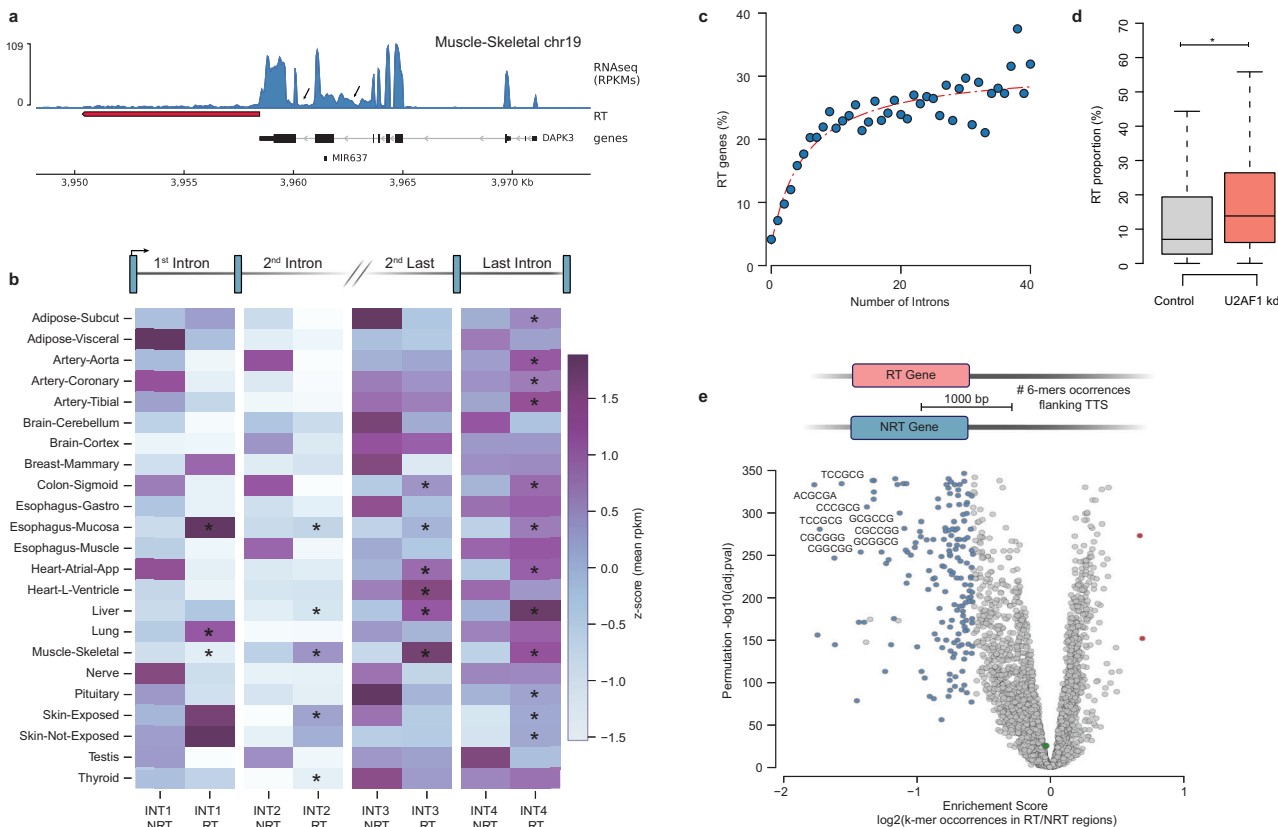

**Fig. 4 Intron retention and Termination signals in RT genes. a** RNAseq profile for the gene *DAPK3* showing intron retention and transcription readthrough region (red) in skeletal muscle. RNAseq coverage is represented as reads per kilobase million (RPKMs). Boxes represent exons separated by introns shown as solid lines. Assembly GRCh38, gencode annotation v37. **b** Heatmap representing the intron retention levels (Z-score of mean RPKMs) in first, second, second -last and last introns of RT and NRT genes of each tissue. Introns where retention was significantly higher in RT vs. NRT are marked with * (Mann-Whitney test *p* value < 0.05 and Choen's d > 0.3). **c** Scatterplot of RT genes (%) and the number of introns for all expressed genes. **d** Readthrough Proportion of RT genes in control and U2AK1 knockout human cells. Boxplot whiskers represent the 25th and 75th percentiles, while the mid-line in each box represents the median. * Mann-Whitney test *p* value < 0.05. **e** Volcano plot representing the enrichment score (log 2 k-mer occurrences in RT and NRT regions) and statistical significance (permutation -log10 *p* value) for the assessed hexamers in the thyroid.

investigate the sequence patterns at the 3' end of RT genes in comparison with NRT genes. We assessed the enrichment of all possible 6-mers in the region immediately upstream and downstream of the TTS (500 bp in each direction) using permutation analysis (Fig. 4e; Methods for details). Our analysis revealed a significant scarcity of GC-rich k-mers in RT genes comparing to NRT genes (e.g. CCGCGC, CCGCGG, CCGUCG CGCCGC) consistently across all tissues (Fig. 4e; Supplementary Data 5). Others have shown that GC content is correlated with a slower elongation time, allowing more time for the 3'-cleavage complex to find the poly(A) site[28]. This depletion of GC clusters at the end of RT genes might create a shorter window of opportunity for the RNA cleavage at the desired specific site, thereby increasing the likelihood of transcription readthrough. Surprisingly, we could not observe the previously described depletion of polyA signals in stress-induced transcription readthrough[5]. Such results suggest that the occurrence of transcription readthrough in healthy tissues is not determined by the presence or strength of the polyA signals.

Therefore, our findings show that failures in co-transcriptional splicing and termination processes may influence transcription readthrough in healthy conditions.

**Transcription readthrough may impact cellular processes**. Our analysis revealed that a large fraction of protein-coding genes could produce RT transcripts in healthy human tissues. However,

it is still unclear how transcription readthrough affects cell homeostasis. Because GTEx samples were profiled using poly(A) enrichment of the mRNA, the depicted RT transcripts may be stable mRNA molecules and possibly translated. Over-representation analysis of the RT genes relative to the expressed genes for each tissue revealed that RT genes are associated with tissue-specific biological processes, such as muscle tissue development in the heart and skeletal muscle regions, steroid binding and monooxygenase activity in the liver, and regulation of the MAPK cascade in adipose tissue (Supplementary Fig. 5a).

The occurrence of transcription readthrough in genes essential for tissue function suggests a possible role for RT transcripts. Thus, we hypothesized that the transcription readthrough could also interfere with protein production through miRNA-mediated gene regulation due to the extension of regions for miRNA binding. More importantly, RT transcripts can act as sponges for miRNAs targeting the respective RT genes. Such a mechanism has been observed for long noncoding RNAs or between pseudogenes and parental genes sharing miRNA binding sites[29,30]. To address this hypothesis, we assessed the miRNA binding sites on the RT tails and the last exon of the respective RT genes in silico. Although the density of binding sites per base pair was not higher (Cohen's d = 0.04), the RT tails contained a higher number of nonbinding sites relative to the last exons (Cohen's d = 0.67 and Mann-Whitney rank test *p* value < 0.05, Supplementary Fig. 5b). More importantly, 1576 RT genes (22%

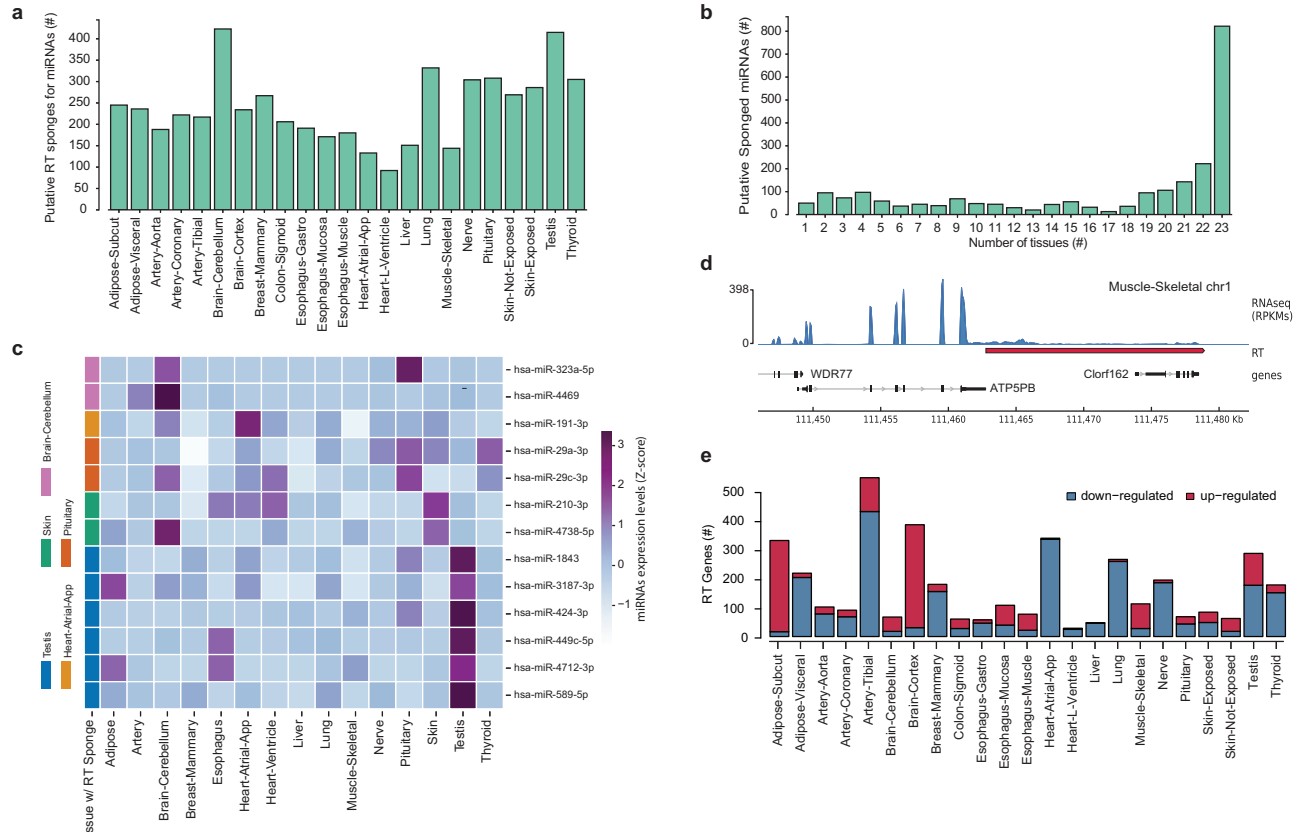

**Fig. 5 Cellular impact of Transcription readthrough. a** Number of predicted RT transcripts acting as putative miRNAs sponges for each tissue (more than 20 binding sites for the same miRNA). **b** Number of tissues containing putative RT sponges for the same miRNA. **c** miRNA expression levels for tissue-specific putative sponged miRNAs (Z-score). **d** RNAseq profile for the gene ATP5PB, showing a read-in example by transcription readthrough (red) in skeletal muscle. RNAseq coverage is represented as reads per kilobase million (RPKMs). Boxes represent exons separated by introns shown as solid lines. Assembly GRCh38, gencode annotation v37. **e** Number of up- or down-regulated RT transcripts in old samples for each human tissue.

of all RT genes) were identified as putative sponges for 2276 miRNAs (85% of total miRNAs analysed) across different tissues (Fig. 5a, Supplementary Fig. 5c, Supplementary Data 6). Most putative sponged miRNAs were common to several tissues, which is consistent with the large number of RT genes shared across tissues (Fig. 5b). Of the 145 miRNAs with putative sponges specific to one or two tissues, we found that 13 were highly expressed in the respective tissue (Fig. 5c and Supplementary Fig. 5d).

We previously showed that cancer-associated TRT events may reach downstream genes that potentially alter their expression levels[4]. Thus, we explored whether the invasion of adjacent downstream genes was frequent in healthy tissues by searching for readthrough tails overlapping the expressed genes downstream. However, only a small fraction of the RT transcripts (<1% across tissues) reached downstream genes (Fig. 5d: Supplementary Fig. 5e). A large fraction of these read-in genes are long noncoding RNAs; however, a few RT transcripts can reach downstream protein-coding genes (Supplementary Fig. 5f).

Finally, given that aging is associated with an increase in transcriptional noise, we presumed that the transcription readthrough could be one of the processes contributing to the production of aberrant RNAs. In fact, older individuals presented higher levels of aberrant transcripts in the cerebral cortex, skeletal muscle, and adipose tissue (Fig. 5e). This indicated that transcription readthrough levels vary with age in a tissue-dependent manner, where they can contribute to cellular fitness loss. In agreement with cellular senescence contributing to individual aging, we also detected higher levels of RT transcripts

in human senescent cells[31] (Supplementary Fig. 5g). Moreover, the higher prevalence of RT transcripts in the cerebral cortex of older adults suggests an association between transcription readthrough and neurodegenerative diseases. Previous transcriptome studies of human cortical regions have revealed increased intron retention associated with progressive aging and Alzheimer's disease[32]. Thus, we hypothesized if inefficient splicing could lead to the production of RT transcripts in brain regions of Alzheimer's patients. In fact, reanalysis of the same human transcriptome profiles[32,33] revealed that the cortical regions of Alzheimer's patients show higher levels of transcription readthrough proportion (Supplementary Fig. 5h).

Therefore, our findings suggest that the production of readthrough transcripts may play a role in competitively sequester of miRNAs and be part of the ageing-related transcription noise.

## Discussion

In recent years, transcription readthrough has emerged as a hallmark of cellular stress responses linked to some human disorders (e.g., cancer and viral infections). However, the prevalence of this phenomenon in healthy human tissues has not yet been explored. Here, we show that readthrough transcription is not exclusive to anomalous conditions and is pervasive across a variety of tissues from healthy individuals. Although conceptually readthrough and alternative polyadenylation may appear equivalent, transcription readthrough has been designated as the extended transcription for several thousand base pairs beyond the annotated gene 3' end[2–5,8]. Thus, such transcripts show a

continuous coverage profile beyond any known isoform or alternative polyadenylation event. Following these criteria, we quantified the transcription readthrough by searching for significant read coverage downstream of all expressed genes in hundreds of transcriptome profiles available from the GTEx project. The use of such a large set of samples allows a larger and more comprehensive portrait of human tissues, but may present some caveats. First, post-mortem changes affect cellular processes such as transcription, impacting transcriptome profiles[21]. To overcome this issue, we selected only adults with fast deaths, and we did not find any association between RT levels and several technical features (e.g., ischemic time and RNA integrity number). Second, the type of RNA-seq libraries available (unstranded and poly A-enriched data) limited our ability to accurately detect all potential RT transcripts. Nevertheless, even after filtering out dubious cases, our analysis revealed that approximately 20% of the expressed genes could undergo transcription readthrough in healthy human tissues. In addition, the correlation found between the RT tail and the respective genes may indicate a lack of sequencing coverage to detect transcription readthrough in genes with low expression. Hence, our underestimated value may represent only the tip of the iceberg regarding transcription readthrough in normal cells. Therefore, only a comprehensive analysis using strand-specific total RNA and augmented coverage will clearly reveal the extension of transcription readthrough in human cells. However, we also detected recurrent production of RT transcripts across tissues, suggesting that transcription readthrough is not a stochastic process.

In fact, our comprehensive analysis identified several distinct features between genes with and without a transcription readthrough in normal cells. We observed that high gene activity restricts the occurrence of transcription readthrough. This finding is consistent with the view that head-on collisions between two converging RNAPIIs may be necessary to prevent transcriptional readthrough[34]. Nevertheless, the expression levels of these genes were insufficient to produce RT transcripts. In agreement with this, other studies have found that RT transcripts are produced regardless of the transcriptional levels of their RT genes in response to stress[5,23].

Furthermore, we found that the production of RT transcripts in healthy tissues was associated with specific epigenetic properties. The TTS vicinity of RT genes showed a higher presence of histone marks typical of transcription elongation (H3K36me3) and accessible regulatory chromatin (H3K4me1 and H3K27ac), consistent with previous observations under conditions of stress or HIV infection[5,6]. However, it is not clear whether alterations in these markers are the cause or consequence of readthrough transcription. More importantly, the ends of the RT tails were associated with the presence of an active enhancer-associated chromatin state, showing higher levels of enhancer-specific histones (H3K4me1, H3K27ac, and H3K4me2) and DNase hypersensitivity profiling. Therefore, our results suggest that RT transcript elongation is restricted by enhancer activity (e.g. binding of multiple proteins or alterations in chromatin conformation).

In addition, our work also discerned the co-occurrence of inefficient splicing and transcription readthrough in healthy human tissues, in agreement with previous studies showing that unspliced mRNAs are typically not cleaved at their 3' ends and often produce readthrough transcripts[35–38]. More importantly, we found a striking correlation between intron number and the likelihood of producing readthrough transcripts, reinforcing that splicing efficiency is an important factor in guiding correct transcription termination. In addition to splicing, RT genes also showed a depletion of GC-rich k-mers in terminal regions. Given that GC-content is associated with a reduction of elongation rate

necessary for 3'-cleavage process (Geisber et al. 2022), we postulate that in RT genes the lack of such specific signals may prevent RNA polymerase stalling and lead to readthrough. Surprisingly, we could not observe a significant depletion of the polyA signal (AAUAAA) as previously associated to stress-induced transcription readthrough (Vilborg et al. 2017). Such results may indicate that the absence or strength of polyA signals is not a major determinant for the occurrence of transcription readthrough in normal cells.

Here, we addressed the potential impact of transcription readthrough in healthy human tissues. According to our results, only a small percentage of readthrough transcripts reach downstream genes, indicating that read-in events are extreme cases of stress or cancer[4]. Nevertheless, we found that RT genes may occur preferentially in genes essential for tissue function, raising questions about their potential role. In fact, we showed that RT transcripts may be putative miRNA sponges, which sequester miRNAs targeting their respective RT genes. Such a role has already been demonstrated for long non-coding RNAs sharing miRNA binding sites with protein-coding genes[30,39]. Moreover, given that intron-containing transcripts have been shown to serve as a source of gene regulation in several contexts[40], one might hypothesize that RT transcripts derived through inefficient splicing may also regulate gene expression via post-transcriptional processing. Finally, we revealed that RT transcripts contribute to the transcription noise that prevails in senescent cells and aging tissues. More importantly, our analysis detected higher levels of aberrantly long RNAs in the cerebral cortex of Alzheimer patients. Although complementary and independent datasets should be explored, our results suggest that transcription readthrough may be prevalent in age-related disorders.

In conclusion, our study revealed that the transcription readthrough is widespread in healthy human tissues, characterized by specific genomic and epigenomic features. Moreover, we unravelled the potential roles of RT transcripts as modulators of gene expression under healthy conditions.

## Methods

**Transcriptome profiles from Human healthy tissues**. RNA samples (BAM files) were accessed on 2021/04/01 from the Genotype-Tissue Expression (GTEx; release v8) project allocated to the NCBI database of Genotypes and Phenotypes (dbGaP)[16–18]. Authorization was granted to dbGaP Accession phs000424.v8. p2, where NIH Genomic Data Sharing Policy policies are applied to protect the privacy of patients (all information is anonymized). The GTEx platform includes approximately 948 postmortem donors, from whom RNA samples from several tissues were isolated in an ongoing manner as donors were enrolled in the study. We considered only paired-end samples with at least 60 million reads per sample and prepared with the Illumina TruSeq library construction protocol (nonstrand specific polyA+ selected library). Cell culture samples and tissues containing fewer than 50 samples were excluded. Healthy subjects were selected by filtering samples for "violent and fast deaths" and "no terminal diseases". We obtained 2778 samples from 23 healthy human tissues that were used for downstream analyses.

**Transcription readthrough detection**. We first converted the downloaded BAM files from dbGaP back to FASTQ using samtools (v.1.10)[41] and re-aligned them to the reference genome (GRCh38 assembly; release 37, GRCh38.p13) using STAR (v2.7.8a)[42]. To detect the transcription readthrough, we used ARTDeco[19], a pipeline for analyzing and characterizing transcriptional readthrough that searches for continuous coverage over a minimal length downstream of the 3'end of each gene

locus (annotation version 37, Ensembl 103) using a rolling window approach. The transcription levels of the window must meet the thresholds to be considered part of the readthrough tail. We used a rolling window of 500 bp, minimum length of 2000 bp, and minimum coverage of 0.15 FPKM. ARTDeco uses HOMER's tools[43] to select only uniquely mapped reads for downstream analysis and returns a variety of metrics to measure readthrough. We used the information contained inside the "quantification" and "dogs" folders (expression levels and novel transcripts created as a result of readthrough, respectively) for downstream analysis.

As GTEx samples were profiled using nonstranded RNAseq libraries, a significant number of reads identified as downstream transcripts corresponded to reads coming from genes being expressed in the opposite direction. Because transcriptional signals can come from either direction, ARTDeco is ambiguous when inferring a true downstream transcript in some cases. To eliminate these dubious cases created by the lack of strandedness (designated as undefined genes), we filtered the output from ARTDeco to report only entries that did not overlap with genes in the opposite strand, using the intersect function from bedtools (v2.30.0)[44]. This approach discards RT transcripts with close downstream neighbors in the opposite strand but ensures that our list of readthrough genes is robust. In addition, only RT transcripts from the expressed genes in each given tissue were considered for downstream analysis. Expressed genes were defined as those with FPKM > 1 in at least 25% of the samples of a given tissue.

To show that some of these ambiguous cases can be either valid transcription readthrough events or false positives, we obtained strand-specific transcriptome profiles[20] from the Gene Expression Omnibus (GEO) repository: adipose tissue (GSM1010958), testis (GSE93500), and heart (GSE93498). The workflow described above was applied to each sample, except for the filter for ambiguous readthrough events. Illustrations of the genomic data were built using the pyGenomeTracks Python module[45].

Hierarchical clustering analysis was performed in Python (Seaborn's Clustermap package) using all 7138 RT genes identified across all tissues. We normalized the median expression of the RT tail across all samples to the respective median expression of the gene body (RTratio = RTlevels/(RTlevels +GeneLevels)). A RT ratio of zero was attributed to RT genes that were not present in certain tissues. Clustering of all expressed genes was performed using the 10000 most variable expressed genes across all tissues and z-score normalization.

**Gene Density Analysis**. To determine the distance between each TTS gene and the closest expressed gene downstream, we used the *closest* function of bedtools (parameters -D a -iu -io –s). For each given gene of interest, bedtools reported the nearest gene found (and the respective distance) from a second input list, containing all expressed genes. Each gene on the first input was labeled as RT or NRT to build the barplot in Supplementary Fig. 2c. We used the Mann-Whitney test to assess statistical significance (p value) and Cohen's d for the effect size.

Chromosome enrichment analysis was performed by counting the number of RT, NRT and expressed genes found in each chromosome and comparison using Fisher's exact test for each chromosome (RT vs. expressed genes; NRT vs expressed genes), adjusting for multiple testing (FDR < 0.05). Enrichment results are displayed as a heatmap representing the detected significance (FDR values) for enrichment (red) and depletion (blue). To estimate the gene density per chromosome, we counted the number of expressed genes divided by each chromosome length. As gene density is not uniform along each chromosome, we performed the same analysis considering regions of 1 megabase

pairs to compute gene density, instead of the whole chromosome. All statistical analyses and plots were produced using built-in functions in the Python environment.

**Epigenetics analysis**. To evaluate whether RT genes had significantly altered epigenetic marks, we used existing chromatin state annotations available from Roadmap Epigenomics and defined by ChromHMM[46]. Under a 25-state ChromHMM model, we considered the following state annotations as active regulatory regions:

1-TssA (Active TSS); 2-PromU (Promoter Upstream TSS); 3-PromD1 (Promoter Downstream TSS 1); 4 - PromD2 (Promoter Downstream TSS 2); 5 - Tx5 (Transcribed - 5' preferential); 6 - Tx (Strong transcription); 7 - Tx3 (Transcribed - 3' preferential); 8 - TxWk (Weak transcription); 9 - TxReg (Transcribed & regulatory (Prom/Enh)); 10 - TxEnh5 (Transcribed 5' preferential and Enh); 11 - TxEnh3 (Transcribed 3' preferential and Enh); 12 - TxEnhW (Transcribed and Weak Enhancer); 13 - EnhA1 (Active Enhancer 1); 14 - EnhA2 (Active Enhancer 2); 15 - EnhAF (Active Enhancer Flank); 16 - EnhW1 (Weak Enhancer 1); 17 - EnhW2 (Weak Enhancer 2); 18 - EnhAc (Primary H3K27ac possible Enhancer); 19 - DNase (Primary DNase); 20 - ZNF/Rpts (ZNF genes & repeats); 21 - Het (Heterochromatin); 22 - PromP (Poised Promoter); 23 - PromBiv (Bivalent Promoter); 24 - ReprPC (Repressed Polycomb); 25 - Quies (Quiescent/Low).

RNAseq samples from nine available tissues were obtained from Roadmap Epigenomics: artery aorta (SRX263858), sigmoid colon (SRX190146), esophagus (SRX190128), left ventricle (SRX190136), liver (SRX218942), lung (SRX190118), psoas muscle (SRX190140), ovary (SRX190120), and thymus (SRX190116). We ran STAR and ARTDeco (as described previously) to assess RT and NRT genes in each sample. We then created bed files containing 2kbp upstream (TTSminus) and downstream of the termination site of each RT and NRT gene (TTSplus) as well as the flanking region (+/−2 kbp) around the end of the readthrough tail. To reduce the overlap of selected regions, we filtered out RT genes with tails shorter than 2500 bp. Finally, we used the ChromHMM OverlapEnrichment module (v1.23) to compute the fold enrichment of each chromatin state along each flanking region.

To further corroborate the enrichment for each chromatin state, we used all available Chip-seq datasets (narrowPeaks) for histone modification marks of these samples from:

https://egg2.wustl.edu/roadmap/data/byFileType/peaks/consolidatedImputed/narrowPeak/

For each dataset, we ran liftOver[47] to convert the hg19 to GRCh38 coordinates. We then computed the number of genes with at least one peak for each chromatin mark in each defined region using bedtools intersect (v2.30.0)[44]. As the RT and NRT groups differed considerably in size and expression, we built 1000 equal-sized expression-matched subsamples by randomly selecting subsets of RT genes from each tissue ($N = 200$) and finding the nearest-neighbor expression partner in the group of NRT genes. We combined the number of RT and NRT regions with or without Chip-seq peaks for a given chromatin mark from all permutations ($N = 1000$), and plotted the average and standard deviation from all subsamples generated. Statistical significance was assessed using the Mann-Whitney U test and Cohen's coefficient.

**Hexamer enrichment analysis**. To test whether RT genes contain different sequence compositions downstream of their termination sites compared to NRT genes, we examined the occurrence of all possible 6-mers in the flanking regions of each TTS

( + / − 500 bp). We defined the enrichment score for each hexamer as the log ratio of the number of occurrences between the two groups.

$$score = \log 2 \left( \frac{\#6mer \; in \; RT \; genes}{\#6mer \; in \; NRT \; genes} \right)$$

To overcome the differences between RT and NRT (size and expression) we applied the same procedure as above, and used 1000 equal-sized expression-matched subsamples ($n = 500$) from each group. To compute the significance of enrichment/depletion of each 6-mer occurrence, we compared the counting distribution obtained from all permutations between the two groups using the $p$ value from a Student's t-test. We applied multiple test correction to the $p$ values and plotted the adjusted values as a function of the enrichment score (volcano plot). Thresholds of 0.001 and 0.58 were applied to the $p$ values and log2ratio values, respectively, to highlight the most enriched (red) and depleted (blue) hexamers.

**Intron retention analysis**. For the intron retention analysis, we built a table (BED format) containing only information concerning introns from all human genes. For this, we first created a matrix containing all regions/coordinates that do not overlap with any exons of any isoform (i.e. introns and intergenic regions), by merging all gene isoforms in the ENSEMBL annotation file (GTF format, GRCh38, version 37, Ensembl 103) and subtracting them from spanning regions of genes using the *complement* module from bedtools (v2.30.0). We then used the bedtools *intersect* module to screen for overlaps between this file and all gene isoforms in the annotation file to generate another BED file containing only intron information (e.g., gene names and coordinates). We filtered this intron table to contain only the first two and last two introns of each gene. Finally, we computed the intron depth coverage for each intron in each sample using bedtool *coverage* and the corresponding RPKM as follows:

$$RPKM = \frac{(ReadCounts + 1)}{intronLength/1000 * TotalNumReads/1e6}$$

Differences in gene expression and group size between RT and NRT genes, was once again handled by randomly picking expression-matched subsamples (500 genes) and run the analysis for multiple subsamples (1000 permutations). Barplots show the average RPKMs values across all samples from all permutations for a given intron (Supplementary Figure 4c). The heatmap shows the median value of each intron contained in the bar plots (Supplementary Figure 4a). To show the cause-effect link between inefficient splicing and transcription readthrough, we used transcriptome profiles from the Encode project before (GSE78686) and after depletion of the splicing factor U2AF1 (GSE88226) in the human cell line (HepG2). Assessment and comparison of transcription readthrough proportions were performed as described above.

**Gene enrichment analysis**. Gene set enrichment analysis for RT genes in each tissue was performed using ToppFun Suite (Chen et al., 2009) based on gene ontology, phenotype, and literature co-citation libraries. The list of expressed genes in each tissue was used as the background for enrichment analysis. For the enrichment test of common RT genes, a list of genes commonly expressed among the tissues was used as the background. The results were considered significant when the adjusted p-values (FDR) were below 0.05.

**Aging and related conditions analysis**. To assess how transcription readthrough varied in aging tissues, we compared the

RT proportion (calculated as described above) of young (<40) and old (>60) individuals for each tissue type using the Mann-Whitney test (FDR < 0.05). RT transcripts were designated as up- or downregulated according to the fold-change of RT proportion between old and young individuals. Senescent cells were characterized using transcriptome profiles for different human cell lines (GSE63577), and early and senescent cells were defined according to a previous study[27]. The comparison of brain regions of healthy individuals and Alzheimer's patients was performed using transcriptome profiles from public cohorts in the Short Read Archive: SRS373308 and SRS373257[33]. Assessment and comparison of transcription readthrough proportions were performed as described above.

**Identification of miRNA targets**. We identified miRNA binding sites for the last exon and tail of the readthrough transcript of the respective RT gene in each tissue. First, we obtained the DNA sequences of each region using the getfasta function of bedtools (v2.30.0). Second, we selected 2650 mature human miRNA sequences obtained from miRBase[48] and used the microRNA BioConductor package[49] (R package version 1.58.0) to reverse complement and obtain the seed regions (starting position 2 and stop position 9). We then used the scanMiR BioConductor package[50] (R package version 1.6.0) to identify miRNA-binding sites in both regions (function findSeedMatches). We calculated the density of the binding sites by dividing them by the respective region length (last exon or RT tail). The comparison between the RT tail and the last exon was performed by averaging the number/density of miRNA binding sites for each RT gene across tissues. Differences were assessed using the Mann-Whitney test and Cohen's d for the effect size. miRNA sponges were defined as RT genes containing at least 20 binding sites for miRNAs, as previously stated[51]. Finally, the miRNA expression profiles for healthy tissues were obtained from microRNA Tissue Expression Database (miTED)[52].

**Statistics and reproducibility**. The statistical methods employed in each analysis are described in their respective sections. All these statistical tests were conducted using dedicated Python packages tailored to each specific analysis.

**Ethical approval**. The collection of biospecimens from deceased individuals is not legally classified as human subjects research; however, sites involved in the GTEx project were mandated to secure written or recorded verbal authorization from the next of kin for the participation of deceased donors. This was typically facilitated through an addendum or modification to an existing authorization form for the donation of tissues and organs for research. The remaining data used in this manuscript (i.e. Roadmap Epigenomics, Senescence and Alzheimer's samples) followed all relevant ethical regulations for work with human subjects and informed consent was obtained from all participants.

**Reporting summary**. Further information on research design is available in the Nature Portfolio Reporting Summary linked to this article.

## Data availability

All main RNA-seq data from human tissues used in this manuscript were accessed through the GTEx-project (release v8) allocated in the NCBI database of Genotypes and Phenotypes (dbGaP), under the dbGAP accession number phs000424.v8. All RNA-seq samples for the study of epigenomic features (Fig. 3) were obtained through the Roadmap Epigenomics platform, containing the following Short Read Archive accession numbers: artery aorta (SRX263858), sigmoid colon (SRX190146), esophagus (SRX190128), left ventricle (SRX190136), liver (SRX218942), lung (SRX190118),

psoas muscle (SRX190140), ovary (SRX190120), and thymus (SRX190116). Senescence and Alzheimer's samples used in Supplementary Figure 5, were obtained from the GEO dataset GSE63577 and from the Short Read Archive SRS373308 and SRS373257, respectively. All the essential data required to replicate the analyses has been made accessible on GitHub, and can be accessed via https://doi.org/10.5281/zenodo.10452611[53]. Source data underlying most figure panels is also available there.

## Code availability

All the code needed to reproduce the analyses has been made accessible on GitHub, and can be accessed via https://doi.org/10.5281/zenodo.10452611[53].

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

## Acknowledgements

We thank João Neto, Silvia Carvalho, and the remaining Computational Multi-Omics Lab members at UCIBIO for their helpful comments and suggestions. Paulo Caldas was a recipient of a Marie Skłodowska-Curie Postdoctoral Fellowship (FOX-MTN-HORIZON-MSCA - 2021-PF-01-01). This work was financed by national funds from Fundação para a Ciência e Tecnologia (FCT), in the scope of the projects UIDP/04378/2020 and UIDB/

04378/2020 of the Research Unit on Applied Molecular Biosciences (UCIBIO) and the project LA/P/0140/2020 of the Associate Laboratory Institute for Health and Bioeconomy - i4HB. The Genotype-Tissue Expression (GTEx) Project was supported by the Common Fund of the Office of the Director of the National Institutes of Health and by NCI, NHGRI, NHLBI, NIDA, NIMH, and NINDS.

## Author contributions

P.C., M.L., S.B., and R.A. conducted bioinformatics analyses. P.C., D.S., and A.R.G. were responsible for critical revision of the manuscript and for important intellectual content. P.C. and A.R.G. conceived and designed the study, interpreted data, and wrote the manuscript. All the authors have read and agreed to the published version of the manuscript.

## Competing interests

Ana Rita Grosso is an Editorial Board Member for *Communications Biology* but was not involved in the editorial review of, nor the decision to publish this article. The other authors declare no competing interests.
