## [Peer Review File · Communications Biology]

Reviewers' comments:

Reviewer #1 (Remarks to the Author):

Review – Caldas et al.

In this manuscript, Caldas et al. present an general overview analysis of the extent of transcriptional readthrough (termed here RT) within healthy tissues, using the GTex dataset. They show that readthrough is much more prevalent than originally appreciated, occurring in overall 9-18% of all transcripts expressed within each tissue, and overall a third of the human tissue transcriptome. They further investigated some features associated with RT, including open chromatin marks, high GC content, R-loops, intron retention, and more. Some of these features are in agreement with previous findings (such as open chromatin and GC content), some have not been examined previously (R-loops), and others have been examined in other scenarios (e.g. stress-induced readthrough) but were found to differ (such as intron retention, read-in). Finally, the authors analyzed age-associated datasets as well as two Alzheimer's disease datasets, and show a slight increase in readthrough with age in a few specific tissues, as well as an increased prevalence of readthrough-positive genes in Alzheimer's disease.

Overall, the topic is timely and interesting, however the analyses presented could have been done more deeply. Often times the results do not relate to previous literature in the field that showed similar findings. It is possible that some of the authors findings are novel, however without direct comparisons and further details one cannot assess what is novel and what is a confirmation of previous findings in other systems.

Currently, this seems like an overview of many features examined, some have been examined previously by others and some were not, however, none of them is further pursued for in-depth analyses to understand their potential causes or consequences. One such example, which is potentially novel, is the "miRNA sponging" hypothesis, that remains as such without any additional investigation. Even the most interesting finding, aging and Alzheimer's, together with the claim that a causal relationship with intron retention might exist in these conditions, is each shown by a single bar plot and not further pursued.

Major comments:

1. A general description of the initial findings is lacking. For example, a clustergram analysis of the patterns of RT in different tissues would be very interesting to see. How do tissues cluster? Is there a pattern? Which tissues are similar? This could really illustrate the tissue specificity as well as housekeeping nature of RTs. RT levels should be normalized to the gene's expression of course.
2. In p.3 the authors show that NRT genes tend to be more lowly expressed. However, in Vilborg et al. 2017 the authors showed no such correlation in the the cells. Could this result merely reflect differenced in the levels of detection? Meaning, that within lowly expressed genes the RT cannot be detected because of sequencing depth? This alternative should be tested.
3. Fig. S2E and the findings that RT genes are further away from neighboring genes are opposing previous studies. It is not clear what is presented in S2E, as the bars stand at ~200kb while previous studies showed distance distributions centered around less than 100kb (Trinklein et al. for example). This should be described in greater detail, and distributions need to be shown in order to make the point.
4. Fig. 2D – this analysis is interesting, especially since previously Vilborg et al. 2017 claimed the opposite. Yet, the analysis as presented is very crude, as within each chromosomes the density of genes is not uniform. It should be performed in higher resolution, specifically for the RT genes within each chromosome.

5. Fig 2C: the variability in the enrichment between different tissues is also intriguing. However, one control is missing: if the same analysis was to be performed for genes, rather than RTs, would the same picture emerge? Meaning, is this result mirroring the gene expression patterns within tissues? Or there is something more to it that hints at some interesting biology? And if so, what happens if this matrix is clustered? Does the tissue clustering make any biological sense?
6. "our findings indicate that increased accessibility favors transcription readthrough." (p.4) – this has been previously shown by others (Vilborg et al. 2017, Hennig et al. 2018, Hadar et al. 2022). Since the actual histone modifications in Fig. 2E-H are not stated it is hard to compare. First, the the actual marks/mark combinations should be directly written in the figure. Second, how do these results compare to single histone mark enrichment analyses? How do they compare to the previously published findings on open chromatin at the RT regions? Do the authors confirm the previous findings in this area or add something new, and if so than what is it?
7. Fig 3A: where is the polyA signal? Table S3 was not provided to the reviewers so I could not look for it there. In any case, if it was not depleted, than the permutation test used to obtain a p-value is perhaps not the correct one to use here. Additionally, in Vilborg et al. the polyT motif was not enriched. What could underlie the difference in results?
8. The results regarding R-loops are intriguing, as they contradict the canonical model claiming that R-loops promote transcription termination. How could this discrepancy be reconciled? Does the position of the R-loops make a difference? Meaning, is it more downstream in the case of RTs?
9. The intron retention results are also interesting, however the analysis was not controlled for initial expression levels, which can confound intron RPKMs, in Fig. 4A, C and S3A and B. what does the trend look like when controlling for this potential confounder?
10. Furthermore, GC content is known as a genomic correlate to intron retention under basal conditions. Can this explain the trend observed for last introns?
11. The functional enrichment analysis should also be controlled for expression levels in each tissue, to rule out high expression as confounder.
12. Has the miRNA analysis been normalized for length? It seems from the figures that it has not. Is the additional number of miRNA sites merely a consequence of the elongated transcript? Or is the density greater as well?
13. Also, are specific miRNAs enriched? Which ones? And are they different between tissues? Does the pattern have anything to do with the tissue expression pattern of the potentially "sponged" miRNA?
14. The authors propose a causal link between intron retention and readthrough in Alzheimer's patients. This should be directly addressed in the Alzheimer's RNA-seq samples, on a gene-by-gene basis, to establish the connection.
15. Moreover, is this connection also true for old individuals? Or just Alzheimer's?
16. Finally, it would be nice and more convincing to show this trend of increased readthrough in Alzheimer's, and the suggested correlation with intron retention, across additional aging and Alzheimer's RNAseq datasets.

Minot comments:

1. The reference to Liang et al. with respect to heat shock and oxidative stress is inaccurate.
2. The manuscript could benefit from linguistic editing.
3. "The number of RT genes varied within each tissue (Fig. Sup. 1C), which could be explained by heterogeneity in tissue samples or sequencing coverage" – sequencing coverage as a confounder could be easily checked.
4. "undefined" genes in fig. 1C and p. 3– how were those defined exactly?

Reviewer #2 (Remarks to the Author):

In this study from Caldas, Grosso and colleagues characterize the readthrough transcription (RT) in the normal samples across several human tissues. They use the GTEx dataset complemented with other datasets. This is a relevant study since the RT phenomenon is poorly characterized in certain conditions.

This paper is well written, easy to follow, with well composed figures. The methods are adequate and state-of-the-art.

I am in principle favourable to the publication of this paper. Nevertheless, I have some questions, which I think, if clarified, could improve some aspects of the work.

1. My main question is how are the processes of RT and alternative polyadenylation (APA) distinguished? In APA, one could have a transcript that includes the use of a more distal polyadenylation site, leading to a longer 3'UTR. Conceptually, one could consider them as equivalent events, but I did not see that discussed in the text. Is there evidence that RT transcripts are polyadenylated?

In Figure 3, it is discussed the over-representation of different signals. What about the PA site signals? One possibility to check for the existence of APA as RT events would be to use poly-A reads, i.e. reads that contain a poly-A tail, that haven't been mapped to the genome. One can trim the poly-A tails, remap them again, cluster and identify APA sites.

2. Members of the GTEx consortium have recently generated long-read RNAseq data from Oxford Nanopore. It would be interesting to see if through these long reads one can also detect events of RT in different samples. In principle, some of these events can be detected with this technology. See:

- <https://gtexportal.org/home/datasets>
- <https://www.biorxiv.org/content/10.1101/2021.01.22.427687v1>

3. Is there any evidence of splicing (eventually called alternative), between an annotated intron/exon and a region in the tail of the RT? Evidence to support this could be obtained by looking at split-maps. The idea is that such events could provide additional evidence of RT.

4. It is indicated that RT tend to be shared in >2 tissues. Since GTEx provides multiple tissue samples per donor, did you find RT events detected in multiple tissues of the same donor? Are the donor specific events? If so, this could potentially indicate support for genetic predisposition, where a SNP in an individual may disrupt a signal that leads to RT. Please discuss.

5. Please explain better the analysis of the cortical region samples. The connection between RT and the Alzheimer disease seems a little bit farfetched and possibly needs stronger support with additional datasets and experiments or clarification.

6. Was RT detected in the cell cultured samples? If so, how does it compare with tissue samples?

7. GTEx samples are obtained as post-mortem samples. The collection and preparation procedures may have an impact, that is tissue dependent. Did the authors investigated the relation of RT with the available technical covariates of GTEx. In particular, variables like RIN, Ischemic time, cause of death and others may have an impact on the expression patterns. Would it be possible to think of a measure of RT levels per sample and from that correlate with the available covariates, eventually including some of the clinical variables? Please discuss.

Minor points:

1. Write GTEx consistently in the text.
2. Beyond the Carithers paper, the authors should also include other GTEx papers, namely:

The Genotype-Tissue Expression (GTEx) project
The GTEx Consortium.
Nature Genetics. 29 May 2013. 45(6):580-5. doi: 10.1038/ng.2653
PMID: 23715323

And for the different phases of the project, the main papers:

The Genotype-Tissue Expression (GTEx) pilot analysis: Multitissue gene regulation in humans
The GTEx Consortium.
Science. 8 May 2015. 348(6235):648-660. doi:10.1126/science
PMID: 25954001

Genetic effects on gene expression across human tissues
The GTEx Consortium.
Nature. 12 Oct 2017. 550: 204-213. Epub 11 Oct 2017. doi:10.1038/nature24277

The GTEx Consortium atlas of genetic regulatory effects across human tissues
The GTEx Consortium.
Science. 369 (1318-1330), 10 Sep 2020. doi:10.1126/science.aaz1776

Reviewer #3 (Remarks to the Author):

1. Brief summary of the manuscript

The study touches on a very interesting and actual problem - readthrough transcription, which is very important for the regulation of gene expression, both in normal and in various pathologies. The authors showed the existence of RT transcripts in healthy tissues genes and suggest that transcription readthrough occurrence may be associated with cellular proliferation rates. However, they did not find confirmation of this. Perhaps the authors should pay attention to the origin of these tissues - the ectoderm gives rise to the nervous tissue and sex glands, and the mesoderm gives rise to the heart and skeletal muscles. It is possible that the patterns of "physiological" RT transcripts (occurring in healthy tissues) are set in early embryogenesis, determining tissue differentiation. It would be interesting to check this in the next work.

The authors confirmed the fact that high levels of gene expression are not sufficient for the formation of RT transcripts and showed that the density of genes in the chromosome is of greater importance (high gene density can restrict the occurrence of transcription readthrough). This fact logically fits with the previously shown fact that a head-on collision between two converging RNAPs is necessary to prevent transcription readout (Wang et al., Mol Cell. 2023 doi: 10.1016/j.molcel.2023.02.017). The authors found that the 3'-end of the RT genes contains a depletion of TA-rich hexamers in the downstream region, a significant enrichment of GC-rich hexamers with the formation of R-loops. In addition, they showed that disruption of the R-loop leads to a decrease in the proportion of RT transcripts. This fact confirms the regulatory properties of R-loops. However, mechanism of this function stays unclear. It was shown that the formation of the R-loop at the 5'end of the gene leads to a decrease in the transcription (D'Souza et al., 2018 doi: 10.1016/j.bbagr.2017.12.008). Possibly,

the formation of R-loops located at the 3' end have the same mechanism of blocking termination with opposite effect on transcription. This opens up new perspectives for research.

The authors found that a large number of introns in expressed genes increases the likelihood of inefficient splicing and the occurrence of RT transcription. This, in turn, can lead to disruption of gene expression through posttranscriptional processing, miRNA-mediated repression, and the accumulation of transcriptional noise. Should such genes be considered as potential causes of aging and disease? Is it possible, on this basis, to isolate specific genes associated with aging, or is it a total process?

2. Overall impression of the work and comments

Clearly, the authors have done a great job of collecting and analyzing a large dataset. The work is interesting and contains a large amount of information that can be used for further research. The importance of the topic suggests that these studies should be of interest to a broad audience interested in genome architecture and gene expression. Unfortunately, the work is not devoid of the shortcomings listed below, without the elimination of which its value is sharply reduced.

1) The authors used the term "transcription readthrough", but there are a huge number of genes with alternative polyadenylation. It must be clearly defined how to distinguish between a gene with an alternative 3'UTR and genes with a transcription readthrough.

2) The authors use post-mortem tissue samples in their work. A large set of samples allows statistically significant conclusions to be drawn. But there is a big problem that in post-mortem samples the expression profile, transcription and processing are changed. There are many publications about this. ((a) The effects of death and post-mortem cold ischemia on human tissue transcriptomes (Ferreira et al., *Nat Commun.* 2018 doi: 10.1038/s41467-017-02772-x). (b) Life and death: A systematic comparison of antemortem and postmortem gene expression (Scott et al., *Gene.* 2020 doi: 10.1016/j.gene.2020.144349)).

There are fears that the observed phenomenon may be associated with such post-mortem changes. The addition of analysis of transcriptomes of biopsy samples obtained during surgical operations would help to solve the problem.

3) Intergenic spaces contain regions of low complexity and repeats. It is not specified whether non-unique reads were filtered out. For genes with low complexity downstream regions transcription readthrough cannot be assessed.

As an example, the downstream regions after the *DYRK1B* and *PSMC4* genes (Figure 1B) have a huge number of similar regions throughout the genome according to BLASTN analysis. In the absence of filtration of non-unique reads, transcription through any of these regions will result in an incorrect level in the downstream regions of these genes.

4) You must specify how the libraries were prepared for sequencing. Is it total RNA or polyadenylated RNA? The answer to this question will allow to correctly interpret the observed transcription readthrough.

RNA polymerase continues transcription after the polyadenylation signal and can be terminated after several thousand base pairs. But due to cutting, the underlying product is not included in the mature product. If RNA is total the detected transcription readthrough may not be part of the transcript and may not affect the final mature transcript. On the contrary, in the samples of polyadenylated RNA, it can indicate alternative polyadenylation.

5) The authors found that the underlying regions of RT genes are enriched with markers of active enhancers. How was it shown that the observed level is determined by the transcription readthrough, and not by the transcription of potential enhancers in these regions?

6) Material and Methods should include a sufficiently detailed description of the procedures to be able to reproduce the described analysis.

7) The second part of title "and determined by inherent genomic features" does not match the content of the article. The authors did not show a causal relationship between genomic features and transcription readthrough.

Point-by Point response to Reviewer #1

In this manuscript, Caldas et al. present an general overview analysis of the extent of transcriptional readthrough (termed here RT) within healthy tissues, using the GTex dataset. They show that readthrough is much more prevalent than originally appreciated, occurring in overall 9-18% of all transcripts expressed within each tissue, and overall a third of the human tissue transcriptome. They further investigated some features associated with RT, including open chromatin marks, high GC content, R-loops, intron retention, and more. Some of these features are in agreement with previous findings (such as open chromatin and GC content), some have not been examined previously (R-loops), and others have been examined in other scenarios (e.g. stress-induced readthrough) but were found to differ (such as intron retention, read-in). Finally, the authors analyzed age-associated datasets as well as two Alzheimer's disease datasets and show a slight increase in readthrough with age in a few specific tissues, as well as an increased prevalence of readthrough-positive genes in Alzheimer's disease.

Overall, the topic is timely and interesting, however the analyses presented could have been done more deeply. Often times the results do not relate to previous literature in the field that showed similar findings. It is possible that some of the authors findings are novel, however without direct comparisons and further details one cannot assess what is novel and what is a confirmation of previous findings in other systems. Currently, this seems like an overview of many features examined, some have been examined previously by others and some were not, however, none of them is further pursued for in-depth analyses to understand their potential causes or consequences. One such example, which is potentially novel, is the “miRNA sponging” hypothesis, that remains as such without any additional investigation. Even the most interesting finding, aging and Alzheimer’s, together with the claim that a causal relationship with intron retention might exist in these conditions, is each shown by a single bar plot and not further pursued.

Major comments:

1. A general description of the initial findings is lacking. For example, a clustergram analysis of the patterns of RT in different tissues would be very interesting to see. How do tissues cluster? Is there a pattern? Which tissues are similar? This could really illustrate the tissue specificity as well as housekeeping nature of RTs. RT levels should be normalized to the gene’s expression of course.

Response: We acknowledge the reviewer for this suggestion. We performed a clustering analysis using RT ratio levels (RT tail levels normalized for gene expression level) that essentially grouped the tissues according to its physiology. A similar clustering was obtained when using expressed genes, suggesting that RT reflects the tissue or cell-specific expression programs. We now added these two panels as Fig. Sup. 1G, re-wrote the last paragraph of the results section accordingly (page 3, line 120) and add the approach in the methods section (page 10, line 423).

2. In p.3 the authors show that NRT genes tend to be more lowly expressed. However, in Vilborg et al. 2017 the authors showed no such correlation in the cells. Could this result merely reflect differences in the levels of detection? Meaning, that within lowly expressed genes the RT cannot be detected because of sequencing depth? This alternative should be tested.

Response: We agree with Vilborg et al. 2017 that expression levels are not the main driver of TRT. Indeed, we pointed out that “more than 70% of highly expressed genes exhibit no transcription readthrough (Fig. Sup. 2C), showing that elevated gene expression is not sufficient to produce RT transcripts”. To clarify this result and overcome the statistical problem caused by large sample sizes, we applied now a stricter

approach reporting also the effect size through Cohen’s coefficient (considering not relevant differences where $d < 0.25$, even for p-values lower than 0.05). Using such approach, we observe that although statistically significant (Mann-Whitney rank test FDR < 0.05), the magnitude of differences between RT and NRT gene expression levels is small (Cohen’s $d < 0.25$). Therefore, we re-wrote the text in the results section (page 4, line 124), rearranging the figures to highlight the most relevant findings in main figure (Fig. 2A -> Fig. Sup. 2A; Fig. Sup. 2A,B -> Fig. Sup. 2B,C; Fig. 2C -> Fig. 2A)

3. Fig. S2E and the findings that RT genes are further away from neighboring genes are opposing previous studies. It is not clear what is presented in S2E, as the bars stand at ~200kb while previous studies showed distance distributions centered around less than 100kb (Trinklein et al. for example). This should be described in greater detail, and distributions need to be shown in order to make the point.

Response: The Figure Sup 2E (new Fig Sup 2C in the revised version) shows all the distances between the end of a given gene and the beginning of the downstream expressed gene in tandem, aiming to assess if the distance to actively transcribed genes would determine RT (expressed genes were defined as genes with FPKM > 1 as described in page 10, line 415 of Methods Section). Such an approach generates longer distances than using all annotated genes and might explain why our results differ from the ones observed by previous studies. Indeed, if considering all annotated genes (instead of only expressed genes), the distance between each RT/NRT gene and the closest-downstream annotated gene in the same strand (tandem) decreases (Figure below). However, following the stricter approach with effect size (Cohen’s d coefficient), we now consider these differences too small to be significant (Cohen’s $d < 0.2$). Thus, we re-wrote the results section to highlight the use of expressed genes and to clarify the small differences found (page 4, line 139).

4. Fig. 2D – this analysis is interesting, especially since previously Vilborg et al. 2017 claimed the opposite. Yet, the analysis as presented is very crude, as within each chromosomes the density of genes is not uniform. It should be performed in higher resolution, specifically for the RT genes within each chromosome.

Response: Like the previous point, the difference between our findings and Vilborg et al 2017 are related to the selection of genes used for the comparison. In Fig 2D (new Fig 2F in revised version), we calculated the gene density using only expressed genes, whereas in previous works the analysis used annotated genes (only filtering out short genes). Given that our focus is to understand the occurrence of RT events and closeness to actively transcribed regions, we believe that we should only consider expressed genes. Nevertheless, to validate our analysis in higher resolution, as suggested by the reviewer, we split each chromosome in smaller regions of 1 mega base pairs and computed expressed gene density (number of expressed genes divided by each region length) and percentage of RT genes (#RT/#EXPGENES) in each region. The analysis with higher resolution shows a similar and significant correlation for all tissues. We included these results as Fig. Sup. 2F, clarify and re-wrote the results (page 4, lines 145) and completed methods section (page 11, line 436).

5. Fig 2C: the variability in the enrichment between different tissues is also intriguing. However, one control is missing: if the same analysis was to be performed for genes, rather than RTs, would the same picture emerge? Meaning, is this result mirroring the gene expression patterns within tissues? Or there is something more to it that hints at some interesting biology? And if so, what happens if this matrix is clustered? Does the tissue clustering make any biological sense?

Response: In Figure 2C we assessed the enrichment of RT genes across the chromosomes comparing to the existence of genes without readthrough in the same region. To address if this is mirroring the gene expression patterns, we now performed two different analyses, comparing: RT genes versus expressed genes; NRT genes versus expressed genes. The novel analyses still show a significant enrichment of RT genes on chromosome 4 and a significant depletion on chromosomes 16 and 17 in several tissues, whereas no enrichment was found for genes without readthrough (Figures below). The clustering analyses grouped some of the tissues according to its physiology. To clarify this point, we have added the new figures and rewritten the text in the Results section (page 4, line 145) and Methods (page 11, line 436).

6. “our findings indicate that increased accessibility favors transcription readthrough.” (p.4) – this has been previously shown by others (Vilborg et al. 2017, Hennig et al. 2018, Hadar et al. 2022). Since the actual histone modifications in Fig. 2E-H are not stated it is hard to compare. First, the the actual marks/mark combinations should be directly written in the figure. Second, how do these results compare to single histone mark enrichment analyses? How do they compare to the previously published findings on open chromatin at the RT regions? Do the authors confirm the previous findings in this area or add something new, and if so than what is it?

Response: We acknowledge the reviewer for this suggestion and have now provided a more complete description of the epigenomic features associated with RT regions, creating a new results subsection (page 4, line 160), completing the Discussion (page 7, 285) and Methods section (page 11, line 446), and add new figures (Fig. 3 and Fig. Sup. 3). We added the chromatin states dictionary indicating the contribution of each histone modification as Fig Sup 3A. Furthermore, we applied a permutation approach with expression-matched genes, similar to Vilborg and co-authors (Vilborg et al 2017), to assess the enrichment of single histone modifications. In fact, when comparing the expression-matched RT and NRT genes, we observed a higher presence of histone marks typical of transcription elongation (H3K36me3) at the 3' end of RT genes relative to genes without readthrough, which continued beyond the transcription termination site (Fig. 3B, C and Sup Fig 3D). Some tissues also showed enrichment of accessible regulatory chromatin (H3K4me1 and H3K27ac) after the transcription termination site (Fig. 3B and Fig Sup D). The presence of these three chromatin marks beyond the 3' end of RT genes has been previously reported under conditions of stress or HIV infection (Vilborg et al. 2017; Henning et al. 2018). Thus, our findings show that such epigenetic alterations are also associated with the production of transcription readthrough in healthy conditions. Furthermore, we also characterized the terminal regions of the RT transcripts, where we found enrichment for an enhancer-associated chromatin state (chromatin state 13; Fig. 3A). Consistently, the ends of RT transcripts showed higher levels of enhancer-specific histones H3K4me1, H3K27ac, and H3K4me2 (Fig 3B, C; Fig Sup 3D). Moreover, we detected high levels of DNase hypersensitivity profiling, indicating increased accessibility. This suggests that enhancer activity may prevent such aberrant elongation.

7. Fig 3A: where is the polyA signal? Table S3 was not provided to the reviewers so I could not look for it there. In any case, if it was not depleted, than the permutation test used to obtain a p-value is perhaps not the correct one to use here. Additionally, in Vilborg et al. the polyT motif was not enriched. What could underlie the difference in results?

Response: We repeated the hexamer analyses using expression-matched gene sets for RT and NRT groups and assessing also the TTS upstream region. Our new analysis revealed only a significant depletion of GC-

rich k-mers in RT genes compared to NRT genes (e.g CCGCGC, CCGCGG, CCGTCG CGCCGC) consistently across all tissues (new Fig. 4D; Sup. Table 4). Others have shown that GC-content is correlated with a slower elongation time, allowing more time for the 3'-cleavage complex to find the poly(A) site (Joseph V Geisberg et al. 2022). This depletion of GC clusters at the end of RT genes might create a shorter window of opportunity for the RNA cleavage at the desired specific site, thereby increasing the likelihood of transcription readthrough. Surprisingly, we could not observe a significant depletion of the polyA signal (AAUAAA) as previously associated to stress-induced transcription readthrough (Vilborg et al 2017). Such results may indicate that the absence or strength of polyA signals is not a major determinant for occurrence of transcription readthrough in normal cells. We provide the new results (page 6, line 213), methods (page 12, line 485) and discussion sections (page 8, line 337).

8. The results regarding R-loops are intriguing, as they contradict the canonical model claiming that R-loops promote transcription termination. How could this discrepancy be reconciled? Does the position of the R-loops make a difference? Meaning, is it more downstream in the case of RTs?

Response: Given that the new reanalysis using expression-matched groups revealed a depletion of GC-rich kmers in the end of RT genes, the original hypothesis of accumulation of R-loops associated with transcription readthrough is not consistently supported. Therefore, we believe that the link between R-loops and transcription readthrough will require new and more analyses that are outside of the scope of our initial goal. Thus, we decided to remove the results regarding R-loops formation in RT genes of the manuscript.

9. The intron retention results are also interesting, however the analysis was not controlled for initial expression levels, which can confound intron RPKMs, in Fig. 4A, C and S3A and B. what does the trend look like when controlling for this potential confounder?

Response: Following the reviewer's suggestion, we repeated the analysis controlling for initial expression levels. When comparing expression-matched genes, we observed intron retention levels higher in RT versus NRT genes for several tissues, but the differences were not consistent across all tissues (new Fig 4A and Fig. Sup. 4 A,B). Thus, we rewrote the results (page 5, line 188) and methods section (page 13, line 514) accordingly.

10. Furthermore, GC content is known as a genomic correlate to intron retention under basal conditions. Can this explain the trend observed for last introns?

Response: We assessed the GC content of the last introns of RT/NRT genes and did not find any significant correlation between: GC content and intron retention levels ($R = -0.06$, $p\text{-value} = 0.17$); or GC content and intron retention levels normalized to gene expression ($R = -0.03$, $p\text{-value} = 0.5$) (Figure below).

11. The functional enrichment analysis should also be controlled for expression levels in each tissue, to rule out high expression as confounder.

Response: The functional enrichment analysis was performed using a list of all expressed genes as background, that was specific of each tissue. This was explained in the Methods sections (page 12, line 508): “The list of expressed genes in each tissue was used as background for each enrichment analysis. For the enrichment test of common RT genes, a list of common expressed genes among tissues was used as background”.

12. Has the miRNA analysis been normalized for length? It seems from the figures that it has not. Is the additional number of miRNA sites merely a consequence of the elongated transcript? Or is the density greater as well?

Response: To normalize for RT tail length and assess the tissue-specificity (point below), we repeated the miRNA analysis per tissue. The RT tails show a higher number of miRNA binding sites relative to the last exons (Cohen’s d = 0.67 and Mann-Whitney rank test p-value < 0.05), but the density is higher in the exons. Given the disparity in lengths of RT tails and last exons, we were not expecting to have higher density. We have added such findings as new Figure Sup 5B, new text in results section (page 6, line 242) and methods section (page 13, line 545).

13. Also, are specific miRNAs enriched? Which ones? And are they different between tissues? Does the pattern have anything to do with the tissue expression pattern of the potentially “sponged” miRNA?

Response: To comprehensively assess the miRNAs sponges we repeated the analysis by tissue and focused on potential miRNAs sponges (RT genes with more than 20 bindings sites for the same miRNA). In total 1576 RT genes (22% of all RT genes) were identified as potential sponges for 2276 miRNAs (85% of total miRNAs) across the different tissues (new Fig 5A, Fig Sup. 5C, Table Sup. 6). Most of the sponged miRNAs were common to several tissues, what is consistent with the large amount of RT genes shared across tissues (new Fig. 5B). From the 145 miRNAs with sponges specific to 1 or 2 tissues, we could find 13 which expression was higher in the respective tissues (Fig. 5C and Fig Sup. 5D). We have added such findings as new Figures 5 and Sup 5, new text in results section (page 6, line 242) and methods section (page 13, line 545).

14. The authors propose a causal link between intron retention and readthrough in Alzheimer’s patients. This should be directly addressed in the Alzheimer’s RNA-seq samples, on a gene-by-gene basis, to establish the connection.

Response: Although we could see a moderate and significant correlation between intron retention and RT levels of the same gene in healthy tissues (Fig Sup 4C), such association was not observed in the Alzheimer’s patient’s samples. As referred above, our goal was to characterize the prevalence and molecular features associated with transcription readthrough in healthy tissues, whereas its occurrence in Alzheimer’s Disease appeared as an additional finding. If the editor and reviewers agree that the current results are not relevant or solid, we will remove them from the paper (i.e. remove the Figure 5F and respective results text).

15. Moreover, is this connection also true for old individuals? Or just Alzheimer’s?

Response: We did not observe an association between intron-levels and transcription readthrough in older individuals. However, we believe that increased production of RT transcripts in ageing may be associated with the loosening of the cell quality checkpoints and consecutive increase in transcription noise.

16. Finally, it would be nice and more convincing to show this trend of increased readthrough in Alzheimer’s, and the suggested correlation with intron retention, across additional aging and Alzheimer’s RNAseq datasets.

Response: Given that the analyses of additional human transcriptome profiles to confirm our findings would require grant access permission, thus we preferred to substitute such results by removing them from the abstract. In fact, our goal was to characterize the prevalence and molecular features associated with

transcription readthrough in healthy tissues, whereas its occurrence in aging and Alzheimer’s Disease appeared as an additional finding. If the editor and reviewers agree that the current results are not relevant or solid, we will remove them from the paper (I.e. remove Fig Sup 5H and respective results text).

Minor comments:

1. The reference to Liang et al. with respect to heat shock and oxidative stress is inaccurate.

Response: We thank the reviewer for this correction, that we altered in page 1 (line 37).

2. The manuscript could benefit from linguistic editing.

Response: We took the comment into consideration and tried to improve the writing.

3. “The number of RT genes varied within each tissue (Fig. Sup. 1C), which could be explained by heterogeneity in tissue samples or sequencing coverage” – sequencing coverage as a confounder could be easily checked.

Response: We did not find any correlation between total read coverage and number of RT genes detected ($R=0.02$, $p\text{-value}=0.8$, Figure below). Such findings suggest that sequencing coverage does not play a major role in the variability of RT genes across samples, thus we removed such confounder from the text (page 3, line 96).

4. “undefined” genes in fig. 1C and p. 3– how were those defined exactly?

Response: We re-wrote the text to clarify how the “undefined” genes were identified in the Results (page 3, line 83) and Methods (page 9, line 380).

Point-by Point response to Reviewer #2

In this study from Caldas, Grosso and colleagues characterize the readthrough transcription (RT) in the normal samples across several human tissues. They use the GTEx dataset complemented with other datasets. This is a relevant study since the RT phenomenon is poorly characterized in certain conditions. This paper is well written, easy to follow, with well composed figures. The methods are adequate and state-of-the-art.

I am in principle favorable to the publication of this paper. Nevertheless, I have some questions, which I think, if clarified, could improve some aspects of the work.

1. My main question is how are the processes of RT and alternative polyadenylation (APA) distinguished? In APA, one could have a transcript that includes the use of a more distal polyadenylation site, leading to a longer 3'UTR. Conceptually, one could consider them as equivalent events, but I did not see that discussed in the text. Is there evidence that RT transcripts are polyadenylated?

In Figure 3, it is discussed the over-representation of different signals. What about the PA site signals? One possibility to check for the existence of APA as RT events would be to use poly-A reads, i.e. reads that contain a poly-A tail, that haven't been mapped to the genome. One can trim the poly-A tails, remap them again, cluster and identify APA sites.

Response: We thank the reviewer for pointing this out. Although conceptually readthrough and alternative polyadenylation may appear equivalent, transcription readthrough has been designated as the extended transcription for several thousand base pairs beyond the annotated gene 3' end (Grosso et al 2015; Vilborg et al 2015; Wisel et al 2018; Rutkowski et al 2015). Thus, such transcripts show a continuous coverage profile beyond any known isoform or alternative polyadenylation event. To clarify this point, we have included this observation in the Discussion section (page 7, line 290). We also emphasize in the methods section that all the RT transcripts found in this work were obtained through polyA-enriched RNAseq (GTEx project). Here, we used polyA-enriched RNAseq profiles and only considered as readthrough regions those with continuous coverage over a minimal length of 2Kbs downstream of the longest annotated isoform (most downstream last exon). Indeed, more than half of the RT transcripts showed an extension of longer than 5 Kbs. To assess the presence of the polyA sites at the end of the RT transcripts, we searched for polyA signal variants around the end of each transcript tail (200 bp up and downstream), In agreement with our hypothesis, most transcripts contained the canonical polyA signal or one of its variants in this region (Figure below).

2. Members of the GTEx consortium have recently generated long-read RNAseq data from Oxford Nanopore. It would be interesting to see if through these long reads one can also detect events of RT in different samples. In principle, some of these events can be detected with this technology.

See:

<https://gtexportal.org/home/datasets>

<https://www.biorxiv.org/content/10.1101/2021.01.22>.

Response: We thank the reviewer for this suggestion. Indeed, using the annotated and novel assembled transcripts generated by the GTEx consortium with long-read RNAseq, we could confirm 911 of the RT transcripts found in the healthy tissues.

3. Is there any evidence of splicing (eventually called alternative), between an annotated intron/exon and a region in the tail of the RT? Evidence to support this could be obtained by looking at split-maps. The idea is that such events could provide additional evidence of RT.

Response: Our previous study in renal cancer showed that RT transcripts reaching downstream genes were subjected to alternative splicing, joining the: second-last exon of the RT gene with the second exon

of the downstream gene (Grosso et al 2015). In healthy tissues, as expected, we found few read-in genes. Regarding alternative splicing events inside the RT gene, as explained above, readthrough regions are characterized by a continuous read coverage beyond gene 3' end, lacking the usual "peaks" associated to exons. Given the short length of sequenced reads/fragments, the split-reads are only found between the end of the last exon and the first nucleotides beyond the gene 3' end.

4. It is indicated that RT tend to be shared in >2 tissues. Since GTEx provides multiple tissue samples per donor, did you find RT events detected in multiple tissues of the same donor? Are the donor specific events? If so, this could potentially indicate support for genetic predisposition, where a SNP in an individual may disrupt a signal that leads to RT. Please discuss.

Response: This is an interesting point. We investigated how often a given RT gene appears in multiple tissues for each donor (normalizing by total number of tissues per donor where the respective gene is expressed). Most RT genes are expressed in less than 25% of the tissues for a given donor (Figure A below), with a small fraction present in all tissues of the same donor. Moreover, when exploring only those RT genes present in all tissues of the same donor (Tissues with RT gene per donor = 100%), we could observe that the majority are also common to more than 200 donors (Figure B below). In fact, we did not find any RT gene to be donor specific events.

5. Please explain better the analysis of the cortical region samples. The connection between RT and the Alzheimer disease seems a little bit farfetched and possibly needs stronger support with additional datasets and experiments or clarification.

Response: The cortical samples were obtained from a previous work in Alzheimer's disease (Adusumalli et al, 2019, Aging Cell, doi: [10.1111/accel.12928](https://doi.org/10.1111/accel.12928)). As we referred above, we realized that the analyses of additional human transcriptome profiles to confirm our findings would require grant access permission. Therefore, we preferred to subside such results by removing them from the abstract. In fact, our goal was to characterize the prevalence and molecular features associated with transcription readthrough in healthy tissues, whereas its occurrence in Alzheimer's Disease appeared as an additional finding. If the

editor and reviewers agree that the current results are not relevant or solid, we will remove them from the paper (i.e. remove the Figure 5F and respective results text).

6. Was RT detected in the cell cultured samples? If so, how does it compare with tissue samples?

Response: The tissue samples showed a higher percentage of expressed genes producing RT transcripts (between 10% and 20%) relative to the cell cultured samples analyzed in the manuscript: 5% in the human cell line (HepG2) and 8% in mouse embryonic stem cells (only controls). We did not include such comparison of tissue versus cell cultures in the manuscript, because it would require the analysis of a wide range of cell lines and our focus was to characterize the production of RT transcripts in healthy tissues.

7. GTEx samples are obtained as post-mortem samples. The collection and preparation procedures may have an impact, that is tissue dependent. Did the authors investigated the relation of RT with the available technical covariates of GTEx. In particular, variables like RIN, Ischemic time, cause of death and others may have an impact on the expression patterns. Would it be possible to think of a measure of RT levels per sample and from that correlate with the available covariates, eventually including some of the clinical variables? Please discuss.

Response: We thank the reviewer for pointing this out. We should emphasize that to secure the quality of the tissue transcriptome profiles, samples were filtered for the same cause of death: “Violent and fast death Deaths due to accident blunt force trauma or suicide (terminal phase estimated at < 10 min.)” or “Fast death of natural causes Sudden unexpected deaths of people who had been reasonably healthy, after a terminal phase estimated at < 1 hr (with sudden death from a myocardial infarction as a model cause of death for this category)”. Thus, this variable should have no effect on our findings.

Furthermore, we investigated the technical variables mentioned by the reviewer (ischemic time, RIN, refrigeration time), and we did not find any significant effect in the number of RT genes detected as a function of those variables across all tissues (Figures below). We also assessed other technical attributes from GTEx and did not find any significant correlation. To clarify this point, we add such findings in the Results section (page 3, line 84), Discussion section (page 7, line 299), and included a new Supplementary Table (Sup. Table 1) containing the correlation coefficients (and adjusted p-values) for several technical and clinical attributes from GTEx samples.

Minor points:

1. Write GTEx consistently in the text.

Response: We thank the reviewer for this correction and have rewritten the GTEx throughout the entire manuscript.

2. Beyond the Carithers paper, the authors should also include other GTEx papers, namely:

The Genotype-Tissue Expression (GTEx) project

The GTEx Consortium.

Nature Genetics. 29 May 2013. 45(6):580-5. doi: 10.1038/ng.2653

PMID: 23715323

And for the different phases of the project, the main papers:

The Genotype-Tissue Expression (GTEx) pilot analysis: Multitissue gene regulation in humans

The GTEx Consortium.

Science. 8 May 2015. 348(6235):648-660. doi:10.1126/science

PMID: 25954001

Genetic effects on gene expression across human tissues

The GTEx Consortium.

Nature. 12 Oct 2017. 550: 204-213. Epub 11 Oct 2017. doi:10.1038/nature24277

The GTEx Consortium atlas of genetic regulatory effects across human tissues

The GTEx Consortium.

Science. 369 (1318-1330), 10 Sep 2020. doi:10.1126/science.aaz1776

Response: We thank the reviewer for this correction and complete the references.

Point-by Point response to Reviewer #3

1. Brief summary of the manuscript

The study touches on a very interesting and actual problem - readthrough transcription, which is very important for the regulation of gene expression, both in normal and in various pathologies. The authors showed the existence of RT transcripts in healthy tissues genes and suggest that transcription readthrough occurrence may be associated with cellular proliferation rates. However, they did not find confirmation of this. Perhaps the authors should pay attention to the origin of these tissues - the ectoderm gives rise to the nervous tissue and sex glands, and the mesoderm gives rise to the heart and skeletal muscles. It is possible that the patterns of "physiological" RT transcripts (occurring in healthy tissues) are set in early embryogenesis, determining tissue differentiation. It would be interesting to check this in the next work.

The authors confirmed the fact that high levels of gene expression are not sufficient for the formation of RT transcripts and showed that the density of genes in the chromosome is of greater importance (high gene density can restrict the occurrence of transcription readthrough). This fact logically fits with the

previously shown fact that a head-on collision between two converging RNAPs is necessary to prevent transcription readout (Wang et al., Mol Cell. 2023 doi: 10.1016/j.molcel.2023.02.017). The authors found that the 3'-end of the RT genes contains a depletion of TA-rich hexamers in the downstream region, a significant enrichment of GC-rich hexamers with the formation of R-loops. In addition, they showed that disruption of the R-loop leads to a decrease in the proportion of RT transcripts. This fact confirms the regulatory properties of R-loops. However, mechanism of this function stays unclear. It was shown that the formation of the R-loop at the 5'end of the gene leads to a decrease in the transcription (D'Souza et al., 2018 doi: 10.1016/j.bbagr.2017.12.008). Possibly, the formation of R-loops located at the 3' end have the same mechanism of blocking termination with opposite effect on transcription. This opens up new perspectives for research.

The authors found that a large number of introns in expressed genes increases the likelihood of inefficient splicing and the occurrence of RT transcription. This, in turn, can lead to disruption of gene expression through posttranscriptional processing, miRNA-mediated repression, and the accumulation of transcriptional noise. Should such genes be considered as potential causes of aging and disease? Is it possible, on this basis, to isolate specific genes associated with aging, or is it a total process?

Response: We would like to thank the reviewer for her/his careful evaluation of our manuscript, appreciating the positive comments and pertinent suggestions. After performing the clustering analysis suggested by Reviewer 1, we assessed if the RT levels clustered the tissues according to their embryonic origin (new Figure Sup. 1). Indeed, the clustering revealed an outlier group composed of ectoderm derived tissues: testis, pituitary and the two brain regions (cortex and cerebellum). However, the two other two major subgroups formed were a mixture of tissues with different origins, such as: liver (endoderm) clustered with muscle and heart regions (mesoderm).

Regarding the RT genes associated with aging, we could identify some RT genes that were previously described as being dysregulated in ageing (data not shown). However, further analyses and functional assays would be necessary to clarify their role in the cellular ageing process.

2. Overall impression of the work and comments

Clearly, the authors have done a great job of collecting and analyzing a large dataset. The work is interesting and contains a large amount of information that can be used for further research. The importance of the topic suggests that these studies should be of interest to a broad audience interested in genome architecture and gene expression. Unfortunately, the work is not devoid of the shortcomings listed below, without the elimination of which its value is sharply reduced.

1) The authors used the term “transcription readthrough”, but there are a huge number of genes with alternative polyadenylation. It must be clearly defined how to distinguish between a gene with an alternative 3'UTR and genes with a transcription readthrough.

Response: This point was also raised by Reviewer 2, and we indicate below how we addressed such concern.

We thank the reviewer for pointing this out. Although conceptually readthrough and alternative polyadenylation may appear equivalent, transcription readthrough has been designated as the extended transcription for several thousand base pairs beyond the annotated gene 3' end (Grosso et al 2015; Vilborg et al 2015; Wisel et al 2018; Rutkowski et al 2015). Thus, such transcripts show a continuous coverage profile beyond any known isoform or alternative polyadenylation event. To clarify this point, we have included this observation in the Discussion section (page 7, line 290). We also emphasize in the methods section that all the RT transcripts found in this work were obtained through polyA-enriched RNAseq (GTEx project). Here, we used polyA-enriched RNAseq profiles and only considered as readthrough regions those with continuous coverage over a minimal length of 2Kbs downstream of the longest annotated isoform (most downstream last exon). Indeed, more than half of the RT transcripts showed an extension of longer than 5 Kbs. To assess the presence of the polyA sites at the end of the RT transcripts, we searched for polyA signal variants around the end of each transcript tail (200 bp up and downstream), In agreement with our hypothesis, most transcripts contained the canonical polyA signal or one of its variants in this region (Figure below).

2) The authors use post-mortem tissue samples in their work. A large set of samples allows statistically significant conclusions to be drawn. But there is a big problem that in post-mortem samples the expression profile, transcription and processing are changed. There are many publications about this. ((a) The effects of death and post-mortem cold ischemia on human tissue transcriptomes (Ferreira et al., Nat Commun. 2018 doi: 10.1038/s41467-017-02772-x). (b) Life and death: A systematic comparison of antemortem and postmortem gene expression (Scott et al., Gene. 2020 doi:10.1016/j.gene.2020.144349)). There are fears that the observed phenomenon may be associated with such post-mortem changes. The addition of analysis of transcriptomes of biopsy samples obtained during surgical operations would help to solve the problem.

Response: This point was also raised by Reviewer 2, and we indicate below how we addressed such concern.

We should emphasize that to secure the quality of the tissue transcriptome profiles, samples were filtered for the same cause of death: “Violent and fast death Deaths due to accident blunt force trauma or suicide (terminal phase estimated at < 10 min.)” or “Fast death of natural causes Sudden unexpected deaths of people who had been reasonably healthy, after a terminal phase estimated at < 1 hr (with sudden death from a myocardial infarction as a model cause of death for this category)”. Thus, this variable should have no effect on our findings.

Furthermore, we investigated the technical variables mentioned by the reviewer (ischemic time, RIN, refrigeration time), and we did not find any significant effect in the number of RT genes detected as a function of those variables across all tissues (Figures below). We also assessed other technical attributes from GTEx and did not find any significant correlation. To clarify this point, we add such findings in the Results section (page 3, line 96), Discussion section (page 8, line 299), and included a new Supplementary Table (Sup. Table 1) containing the correlation coefficients (and adjusted p-values) for several technical and clinical attributes from GTEx samples.

3) Intergenic spaces contain regions of low complexity and repeats. It is not specified whether non-unique reads were filtered out. For genes with low complexity downstream regions transcription readthrough cannot be assessed.

As an example, the downstream regions after the *DYRK1B* and *PSMC4* genes (Figure 1B) have a huge number of similar regions throughout the genome according to BLASTN analysis. In the absence of filtration of non-unique reads, transcription through any of these regions will result in an incorrect level in the downstream regions of these genes.

Response: We agree with the reviewer's point, that allowing multimapping reads could lead to misleading results. In fact, we detected RT transcripts using ARTDeco pipeline, where only uniquely mapped reads are kept for downstream analysis with HOMER's tools (description here: <http://homer.ucsd.edu/homer/ngs/tagDir.html>). To clarify this point, we have emphasized such details in the Methods section (page 10, line 400).

4) You must specify how the libraries were prepared for sequencing. Is it total RNA or polyadenylated RNA? The answer to this question will allow to correctly interpret the observed transcription readthrough. RNA polymerase continues transcription after the polyadenylation signal and can be terminated after several thousand base pairs. But due to cutting, the underlying product is not included in the mature product. If RNA is total the detected transcription readthrough may not be part of the transcript and may not affect the final mature transcript. On the contrary, in the samples of polyadenylated RNA, it can indicate alternative polyadenylation.

Response: We thank the reviewer for pointing this out. The GTEx samples were profiled using the Illumina TruSeq library construction protocol (non-strand specific polyA+ selected library), reinforcing that the RT transcripts are polyadenylated. We add the information about the library preparation in the Methods section (page 9, line 383). Regarding the difference between alternative polyadenylation and transcription readthrough, we already addressed this in point 1.

5) The authors found that the underlying regions of RT genes are enriched with markers of active enhancers. How was it shown that the observed level is determined by the transcription readthrough, and not by the transcription of potential enhancers in these regions?

Response: We think that these do not correspond to transcription of potential enhancers due to several reasons:

- 1) We observed an enrichment of the chromatin state associated with active enhancer regions at the end of the RT transcript. In fact, the enhancer-specific histones H3K4me1 and H3K27ac only appeared significantly increased in RT genes of some tissues.
- 2) Our analysis with strand-specific RNAseq profiles confirms the direction of the transcription from the gene end and towards the downstream region enriched with chromatin marks of enhancers.
- 3) Our RT transcripts are longer than the usual length associated with eRNAs.
- 4) The majority of the eRNAs described are not polyadenylated and the GTEx transcriptome profiles were obtained by polyA RNA-seq approach.

6) Material and Methods should include a sufficiently detailed description of the procedures to be able to reproduce the described analysis.

Response: We understand the reviewer's suggestions and have rewritten the methods. Furthermore, we have added the subsection "Reproducibility" in the Methods referring: The code and processed data needed to reproduce the analyses and figures are available on the GitHub webpage: <https://github.com/comicsfct/rtHealthyTissues>.

7) The second part of title "and determined by inherent genomic features" does not match the content of the article. The authors did not show a causal relationship between genomic features and transcription readthrough.

Response: We agree with the reviewer's point, thus we altered the title to: "Transcription readthrough is prevalent in healthy human tissues and associated with inherent genomic features"

Reviewers' comments:

Reviewer #2 (Remarks to the Author):

The authors have done an extensive revision of the first manuscript, doing additional analysis to support their claims. They also adapted the text regarding the association of RT to Alzheimers's. The images are carefully presented.

Two minor notes:

- When the GTEx dataset is mentioned, authors should mostly cite the 2015, 2017 and 2020 GTEx Consortium papers as these are the ones that best describe the used dataset.
- Lines 324, 336, Furthermore repeated.

Reviewer #3 (Remarks to the Author):

Clarifying key aspects in the new revision definitely improved the understanding of the article and as a result, new comments and questions arose.

1) The authors contradict themselves:

Initially, the authors introduce the definition that "the transcription machinery fails to identify the termination site and continues transcribing beyond gene boundaries in cancer cells, a phenomenon designated as transcription readthrough." From this definition, it follows that RT is a pathological process associated with a disruption in the polyadenylation process and transcription termination. Later in the text, the authors write, "RT transcripts are typically described as RNA molecules extending at least 5 kbps past the normal termination site of their gene of origin (RT gene) and are retained in the nucleus close to their transcription site, most likely associated with chromatin." By stating this, the authors clearly establish characteristics of what they mean by RT. This leads the reader to think that RT transcripts are a consequence of transcription apparatus malfunction. And since they are retained in the nucleus, they are likely non-polyadenylated.

But the authors write, "The GTEx samples were profiled using the Illumina TruSeq library construction protocol (non-strand specific polyA+ selected library), reinforcing that the RT transcripts are polyadenylated." Firstly, this means that if genes had RT, it wouldn't be detected in the poly(A)-enriched library. Secondly, as a consequence, the transcripts considered in the study underwent normal processing. The authors conducted a search for potential polyadenylation signals at the ends of "RT" and demonstrated that these regions contain canonical poly(A) signals. Thus, alternative isoforms were found for a large number of genes that were previously unidentified or were discarded due to low transcription levels in downstream regions. It is likely that the authors should shift the focus from RT to alternative processing of the 3'-end. Why use a term that characterizes a pathological process of termination for normally processed transcripts in healthy human tissues?

2) The most important characteristic is the sample collection time. The study used samples with "terminal phase estimated at < 10 min" and "a terminal phase estimated at < 1 hr," which is acceptable.

3-7) No objections, corrections have been made.

It would be interesting to see the distribution of the ratios of RT zone read density to the genes producing them.

Reviewer #4 (Remarks to the Author):

The authors of the manuscript "Transcription readthrough is prevalent in healthy human tissues and associated with inherent genomic features" have addressed all the concerns raised in the first round of review.

I agree with the largely consistent opinion that the authors conducted a thorough analysis of an important phenomenon - transcription readthrough - that is of immediate interest to a broader audience.

Despite that, I have remaining concerns regarding the methods used for intron retention and miRNA sponging analysis.

1. The authors have used RPKM as a measure of intron retention. While that isn't an issue, I would recommend using one of the multiple tools available for intron retention analysis (or more generally AS), such as IRFinder, Suppa2, or SpliceWiz that look beyond intron expression (RPKM) and apply specific inclusion/exclusion criteria for intron retention, e.g. evenness of intron coverage, PSI or IR-ratio, etc.

2. Though a higher density of miRNA seed complementary sites were found in RT vs NRT genes I would suggest to use the terms "miRNA sponge" or "sponged miRNAs" with caution. Unless a sponging effect is proven these can merely be referred to as "putative miRNA sponges" or better "putative miRNA target hubs". miRNA target site predictions typically include a conservation analysis and minimum free energy calculation. Some also look for negative correlation of miRNA and gene expression. Again, multiple tools are available that included these analyses.

RESPONSE LETTER

Transcription readthrough is prevalent in healthy human tissues and associated with inherent genomic features.

We thank the Referees for constructive comments and suggestions provided. In response to their valuable feedback, we have addressed each comment in a point-by-point fashion. As before, all alterations made in the manuscript are clearly highlighted in red for easy identification.

Point-by Point response to Reviewer #2

The authors have done an extensive revision of the first manuscript, doing additional analysis to support their claims. They also adapted the text regarding the association of RT to Alzheimers's. The images are carefully presented. Two minor notes:

- When the GTEx dataset is mentioned, authors should mostly cite the 2015, 2017 and 2020 GTEx Consortium papers as these are the ones that best describe the used dataset.
- Lines 324, 336, Furthermore repeated.

Response: We thank the reviewer for the positive comments after we revised the manuscript. We have rectified the repetition of the word "Furthermore" and completed the references suggested.

Point-by Point response to Reviewer #3

Clarifying key aspects in the new revision definitely improved the understanding of the article and as a result, new comments and questions arose.

1) The authors contradict themselves:

Initially, the authors introduce the definition that "the transcription machinery fails to identify the termination site and continues transcribing beyond gene boundaries in cancer cells, a phenomenon designated as transcription readthrough." From this definition, it follows that RT is a pathological process associated with a disruption in the polyadenylation process and transcription termination.

Later in the text, the authors write, "RT transcripts are typically described as RNA molecules extending at least 5 kbps past the normal termination site of their gene of origin (RT gene) and are retained in the nucleus close to their transcription site, most likely associated with chromatin." By stating this, the authors clearly establish characteristics of what they mean by RT. This leads the reader to think that RT transcripts are a consequence of transcription apparatus malfunction. And since they are retained in the nucleus, they are likely non-polyadenylated.

But the authors write, "The GTEx samples were profiled using the Illumina TruSeq library construction protocol (non-strand specific polyA+ selected library), reinforcing that the RT

transcripts are polyadenylated." Firstly, this means that if genes had RT, it wouldn't be detected in the poly(A)-enriched library. Secondly, as a consequence, the transcripts considered in the study underwent normal processing. The authors conducted a search for potential polyadenylation signals at the ends of "RT" and demonstrated that these regions contain canonical poly(A) signals. Thus, alternative isoforms were found for a large number of genes that were previously unidentified or were discarded due to low transcription levels in downstream regions. It is likely that the authors should shift the focus from RT to alternative processing of the 3'-end. Why use a term that characterizes a pathological process of termination for normally processed transcripts in healthy human tissues?

Response: We thank the reviewer for recognising the improvements we made in our revised manuscript and for highlighting a potential confusion between alternative polyadenylation and transcription readthrough (TRT).

While it is true that TRT has traditionally been characterised as a phenomenon linked to pathogenic conditions (e.g. cellular stress, virus infection, and cancer), others have identified several hundreds of readthrough transcripts in normal/healthy human cells (Vilborg et al., 2015, 2017). Thus, we are not introducing any new definitions but rather building upon established frameworks to explore this phenomenon in human healthy tissues. Furthermore, readthrough transcripts are not simply extremely long 3' UTR extensions. They extend more than 2000 bp beyond the TTS (and can be over 200 kb long) and exhibit a distinct coverage profile than any alternative polyadenylation isoform (Rutkowski et al., 2015; Vilborg et al., 2015, 2017; Hennig et al., 2018).

In addition, others have shown that readthrough transcripts exist in both polyadenylated and non-polyadenylated forms (Vilborg et al., 2015, 2017; Rodríguez-Molina et al., 2023). Also, while mostly non-polyadenylated transcripts are retained in the nucleus, there are exceptions. Some polyadenylated transcripts can also be retained in the nucleus for various regulatory or structural reasons (Bahar Halpern et al., 2015), which is indeed a potential role for TRT transcripts, maintaining chromatin integrity (Vilborg et al., 2015, 2017). The fact that the "GTEx samples were profiled using the Illumina TruSeq library construction protocol" corroborates that we are indeed characterizing polyadenylated downstream-of-gene (DoG)-containing transcripts in this work.

To clarify even further all these observations, we have restructured the introduction and highlighted the first paragraph of the discussion.

2) The most important characteristic is the sample collection time. The study used samples with "terminal phase estimated at < 10 min" and "a terminal phase estimated at < 1 hr," which is acceptable.

3-7) No objections, corrections have been made. It would be interesting to see the distribution of the ratios of RT zone read density to the genes producing them.

Response: We thank the reviewer for recognising the improvements we made in our revised manuscript

Point-by Point response to Reviewer #4

The authors of the manuscript "Transcription readthrough is prevalent in healthy human tissues and associated with inherent genomic features" have addressed all the concerns raised in the first round of review.

I agree with the largely consistent opinion that the authors conducted a thorough analysis of an important phenomenon - transcription readthrough - that is of immediate interest to a broader audience. Despite that, I have remaining concerns regarding the methods used for intron retention and miRNA sponging analysis.

1. The authors have used RPKM as a measure of intron retention. While that isn't an issue, I would recommend using one of the multiple tools available for intron retention analysis (or more generally AS), such as IRFinder, Suppa2, or SpliceWiz that look beyond intron expression (RPKM) and apply specific inclusion/exclusion criteria for intron retention, e.g. evenness of intron coverage, PSI or IR-ratio, etc.

Response: We would like to thank the reviewer for the overall positive evaluation of our manuscript. Regarding the intron retention analysis, our initial strategy involved running Vast-tools for the identification and quantification of alternative splicing (AS) events. However, we encountered two main limitations: 1) Vast-tools directly aligns RNA-seq reads to predetermined sets of exon-exon and exon-intron junctions (EEJs), thereby offering splicing quantification limited to a predefined and unalterable set of AS events (referred to as VASTDB libraries). Consequently, the tool does not provide the desired measure of intron retention for every first and last exon of all expressed genes, as we intended. 2) Vast-tools, as well as other tools of reference, require a large number of reads aligning to the EEJs to discern AS events as significant. Given that our study involves only healthy human tissues (no abnormal perturbation), the number of reads aligning to such regions falls below the standard quality detection threshold (we tested this with Vast-tools). Other tools such as IRFinder do not use such predefined AS event libraries as reference, but they also quantify intron retention by comparing the number of reads spanning exon-intron junctions to the total number of reads covering the corresponding exon-exon junctions. These are great tools to investigate how intron retention events change in response to some cellular perturbation. However, for a broader analysis of overall transcript expression, including introns without a specific focus on splicing events, we found that a simple RPKM quantification was better suited and more computationally efficient. In addition, performing such analysis now would require a substantial time investment, taking into account our dataset comprising 3000 samples. Regardless, our most prominent finding is the correlation between the occurrence of transcription readthrough and the number of introns in all expressed genes, which is independent of the measured levels of intron retention.

2. Though a higher density of miRNA seed complementary sites were found in RT vs NRT genes I would suggest to use the terms "miRNA sponge" or "sponged miRNAs" with caution. Unless a sponging effect is proven these can merely be referred to as "putative miRNA sponges" or better "putative miRNA target hubs". miRNA target site predictions typically include a conservation analysis and minimum free energy calculation. Some also look for negative correlation of miRNA and gene expression. Again, multiple tools are available that included these analyses.

Response: We followed the reviewer's suggestions and replaced the terms with "putative miRNA sponges" and "putative sponged miRNAs". Regarding the methods, we had initially pondered using miRNA target prediction tools including free energy calculations (and other parameters). However, most of such tools use as input the gene name or just accept one sequence at once. In addition, we failed to install and run some of the multiple-sequence tools: TargetScan (the batch version does not provide the folding energy); miRanda (deprecated); miRmap (not possible to install the dependencies); ComiR (deprecated). Given that we needed to predict the binding sites for more than 1000 transcripts, we decided to use the scanMiR BioConductor package. Nevertheless, we would be grateful if the reviewer has any suggestion for a tool that can predict automatically the binding sites of several miRNAs on several RNA sequences.